# Technical note: Assessment of float-pH data quality control methods - A study case in the subpolar northwest Atlantic Ocean

Cathy Wimart-Rousseau[1], Tobias Steinhoff[1], Birgit Klein[3], Henry Bittig[4], and Arne Körtzinger[1,2]

[1]GEOMAR Helmholtz Centre for Ocean Research Kiel, Kiel, Germany
[2]Kiel University, Kiel, Germany
[3]Federal Maritime and Hydrographic Agency (BSH), Hamburg, Germany
[4]Leibniz Institute for Baltic Sea Research Warnemuende (IOW), Seestrasse 15, 18119 Rostock, Germany

**Correspondence:** Cathy Wimart-Rousseau (cwimart-rousseau@geomar.de)

**Abstract.** Since a pH sensor has become available that is principally suitable for use on demanding autonomous measurement platforms, the marine $CO_2$ system can be observed independently and continuously by BGC-Argo floats. This opens the potential to detect variability and long-term changes in interior ocean inorganic carbon storage and quantify the ocean sink for atmospheric $CO_2$. In combination with a second parameter of the marine $CO_2$ system, pH can be a useful tool to derive the surface ocean $CO_2$ partial pressure ($pCO_2$). The large spatiotemporal variability of the marine $CO_2$ system requires sustained observations to decipher trends and study the impacts of short-term events (e.g., eddies, storms, phytoplankton blooms) but also puts a high emphasis on the quality control of float-based pH measurements. In consequence, a consistent and rigorous quality control procedure is being established to correct sensor offsets or drifts as the interpretation of changes depends on accurate data. By applying current standardized routines of the Argo data management to pH measurements from a pH/$O_2$ float pilot array in the subpolar North Atlantic Ocean, we assess the uncertainties and lack of objective criteria associated with the standardized routines, notably the choice of the reference method for the pH correction (CANYON-B, LIR-pH, ESPER-NN and ESPER-LIR) as well the reference depth for this adjustment. For the studied float array, significant differences ranging between ca. 0.003 pH units and ca. 0.04 pH units are observed between the four reference methods which have been proposed to correct float-pH data. Through comparison against discrete and underway pH data from other platforms, an assessment of the adjusted float-pH data quality is presented. The results point out noticeable discrepancies near the surface of > 0.004 pH units. In the context of converting surface ocean pH measurements into $pCO_2$ data for the purpose to derive air-sea $CO_2$ fluxes, we conclude that an accuracy requirement of 0.01 pH units (equivalent to a $pCO_2$ accuracy of 10 $\mu$atm as minimum requirement for potential future inclusion into the SOCAT database) is not systematically achieved in the upper ocean.

While the limited dataset and regional focus of our study does not allow for firm conclusions, the evidence presented still calls for the inclusion of an additional independent pH reference in the surface ocean into the quality control routines. We therefore propose a way forward to enhance the float-pH quality control procedure. In our analysis, the current philosophy of pH data correction against climatological reference data at one single depth in the deep ocean appears insufficient to assure adequate data quality in the surface ocean. Ideally, an additional reference point should be taken at or near the surface where the resulting $pCO_2$ data are of the highest importance to monitor the air-sea exchange of $CO_2$ and would have the potential to very significantly augment the impact of the current observation network.

# 1 Introduction

Since the beginning of the industrial era, the ocean has played a critical role by absorbing about 25% (Friedlingstein et al., 2023) of the annual anthropogenic $CO_2$ emissions, thereby mitigating the current climate change (IPCC, 2021). Ocean $CO_2$ uptake causes changes in the ocean chemistry, inducing an increase in hydronium ion concentration (i.e., a decrease in oceanic pH). Throughout the world ocean, these changes, also termed "ocean acidification" (OA; Doney et al., 2009), are already observed and a global surface ocean pH decline of 0.1 units since the beginning of the industrial era has been reported (Orr et al., 2005). Depending on emission scenarios, ocean acidity will increase with a projected pH decline ranging from 0.16 to 0.44 pH units by 2100 (e.g., Kwiatkowski et al., 2020). These changes, while being variable regionally and along the water column (Carstensen and Duarte, 2019; Orr et al., 2005), represent a significant environmental change and potential threat to marine organisms and marine ecosystems that needs to be elucidated.

To assess long-term changes in ocean chemistry, oceanographic cruises were conducted and discrete water samples were collected. These historical hydrographic data have been synthesized in databases such as the Global Ocean Data Analysis Project (GLODAPv2) database (Olsen et al., 2016) which provides an internally consistent reference data product. However, in addition to anthropogenic modifications, oceanic pH is a dynamic variable in response to biological, physical, and chemical processes and changes on daily to centennial timescales, with pronounced seasonal, interannual, and decadal variability. In consequence, ship-based observing strategies, being often skewed towards certain months and regions, especially in some places where current sampling methods are not possible (e.g., permanently or seasonally ice-covered regions), cannot adequately capture the dynamic spatiotemporal variability of this carbonate system parameter.

In order to improve our understanding of the oceanic $CO_2$ cycle and to decipher any temporal change, sustained time-series measurements at fixed stations have been carried out over the last decades (e.g., Bates et al., 2014). Nevertheless, the low spatial coverage associated with these sampling sites, generally located near coastal areas, precludes a rigorous description of the open-ocean variability. Thus, these long-term data collections, with uneven regional distribution and typically moderate temporal resolutions (i.e., bi-weekly or monthly), lead to "observational gaps" with an under-sampling of biogeochemical variables (Tanhua et al., 2019). Since the 1990s, the Ship Of Opportunity Program (SOOP; Goni et al., 2010) aims to obtain data from autonomous instrumentation installed on volunteer merchant ships regularly crossing certain areas. This network contributes to building sustained carbon observing datasets and complements the limited capacity of classical observational strategies as the standard-SOOP framework features, at least, routine $p$CO_2 observations (e.g., Lüger et al., 2004). In the Atlantic Ocean, parts of the SOOP network are operated in the European Research Infrastructure 'Integrated Carbon Observation System' (ICOS) and the 'Surface Ocean $CO_2$ Reference Observing Network' (SOCONET).

To circumvent these gaps and overcome the existing severe limitations in terms of both spatial and temporal resolutions, autonomous platforms such as moorings, profiling floats, underwater gliders, or surface vehicles have been deployed at a global scale (Bushinsky et al., 2019; Whitt et al., 2020) and contributed to the extension of databases (Abram et al., 2019). Recently, the development of a pH sensor suitable for deployment on autonomous platforms has extended our observation capabilities of the marine $CO_2$ system (Johnson et al., 2016)

Defined as an Essential Ocean Variable (EOV) by the Global Ocean Observing System (GOOS, www.goosocean.org), pH can be used to determine marine $CO_2$ system changes in response to anthropogenic impacts. However, the key to this autonomous platform expansion is the achievable and documented quality of the pH data which relies on defined practices ranging from rigorous pre-deployment sensor calibration to post-deployment assurance of data accuracy and consistency (Johnson et al., 2018). Indeed, for reliably identifying and interpreting change, accurate and consistent data are needed.

For BGC-Argo floats data, operational procedures for physical data (temperature, salinity, pressure) quality control (QC) have been established, ranging from automated "Real-Time" (RT) checks to sophisticated "Delayed-Mode" (DM) adjustments (Schmechtig et al., 2016; Wong et al., 2022). For pH, numerous delayed-mode procedures have been suggested (Williams et al., 2016; Johnson et al., 2017) but a uniform, fully tested and globally-proven correction method is still missing. Recently, in the framework of the Southern Ocean Carbon and Climate Observations and Modelling project (SOCCOM; Russell et al.,
2014), a methodology has been developed to correct nitrate, pH, and oxygen values from sensor drifts and offsets in DM. Two Matlab tools named SAGE (SOCCOM Assessment and Graphical Evaluation) and SAGE-O2 have been created as interfaces to support the validation and correction of float-pH and oxygen data, respectively. In the SAGE procedure (Maurer et al., 2021), the machine learning method 'Carbonate system and Nutrient concentration from hYdrological properties and Oxygen, Bayesian approach' (CANYON-B; Bittig et al., 2018b), the Locally Interpolated Regression (LIR) algorithmic method (Carter
et al., 2018) and multiple linear regression techniques (Williams et al., 2016) are used as a reference to correct float-pH data at depths of typically around 1500 dbar. The neural-network CANYON-B approach is based on the approach originally developed by Sauzède et al. (2017). Recently, two Empirical Seawater Property Estimation Routines (ESPER; Carter et al., 2021) have been included in SAGE as reference methods. The ESPER-NN method generates estimates from neural networks while the ESPER-LIR routine is based on locally interpolated regressions.

In this study, we have used the SAGE tool and the included adjustment methods to correct float-pH data acquired from a pilot array established in 2018 in the subpolar northwest Atlantic Ocean (SNWA), a region of particular relevance in the marine carbon cycle. This area is a key region for anthropogenic carbon uptake and storage (Sabine et al., 2004; Gruber et al., 2009; Khatiwala et al., 2013; Racapé et al., 2018) as a consequence of (1) the Meridional Overturning Circulation (MOC) transporting warm and anthropogenic carbon-laden tropical waters by its upper limb (Sabine et al., 2004; Gruber et al., 2009; Khatiwala et
al., 2013), and to (2) deep winter convection events occurring in the Labrador and Irminger Seas which transfer anthropogenic carbon from surface to the deep ocean (Körtzinger et al., 1999; Sabine et al., 2004; Ridge and McKinley, 2020). Moreover, it should be noted that the North Atlantic Oscillation (NAO), through its impact on the atmospheric variability in the North Atlantic region, induces high temporal variability on interannual (Watson et al., 2009) to decadal time scales (Leseurre et al., 2020) and may alter the residence time of anthropogenic carbon in the ocean by altering the rate of water mass transformation
(Levine et al., 2011). In this context, the study region can be considered both a region of highest interest and a region of methodological challenges.

This paper illustrates the performance of the proposed standard Argo quality control routines with the float-pH data acquired in the SNWA region. By using float-pH data and independent pH data measured from water samples collected at a nearby station as well as underway data obtained from an autonomous platform in the SNWA area, we can provide an evaluation of

95 (1) the impact of the choice of the at-depth reference pressure as well as the choice of the reference method used to correct float-pH data, (2) differences to co-located in situ discrete pH data over the water column and within the surface layer and, (3) differences to crossovers to in situ surface pH data collected along a ship-of-opportunity line.

## 2   Materials and Methods

### 2.1   BGC-Argo float array

As part of an ongoing pilot study, 10 BGC-Argo floats from two manufacturers (NKE instrumentation and Teledyne Webb Research) were deployed in the SNWA region (Fig. 1) over the 2018 to 2022 period. All floats were equipped with pressure, temperature, salinity (SBE-41CP sensor, Sea-Bird Electronics), oxygen (oxygen optode 4330 with individual multi-point manufacturer calibration, Aanderaa Data Instruments), and pH sensors (SeaFET$^{TM}$ sensor, Sea-Bird Electronics, Inc.). As some of the BGC-Argo floats considered here are still operational, no DM data are available yet for the entire dataset. BGC-Argo
data were obtained from the Coriolis Data Assembly Center. For inactive BGC-Argo floats, the Argo real-time quality control procedures have been applied by the Coriolis data center (Wong et al., 2022). Temperature and salinity measurements (derived from conductivity) are recorded with accuracies of $\pm$ 0.002 °C and $\pm$ 0.005 PSU. The initial pH accuracy "claimed" by the manufacturer is of $\pm$ 0.05 pH units. Data adjustment have been reported to yield accuracies varying between $\pm$ 0.005 pH units (Johnson et al., 2017) and $\pm$ 0.007 pH units (Maurer et al., 2021). Oxygen optodes, similar to other chemical sensors,
are known to suffer from storage drift prior to deployment (Bittig and Körtzinger, 2015; Johnson et al., 2015). SAGE-O2, or an equivalent script, must therefore be used to correct float-oxygen data prior to any float-pH data correction which employs oxygen values as ancillary data (e.g., CANYON-B). In this study, oxygen data were used as predictor variables in all reference algorithms used. We note that the oxygen data correction employs in-air measurements routinely carried out during each float surfacing to achieve highest data accuracy (Bittig and Körtzinger, 2015; Bittig et al., 2018a). A stringent referencing and ad-
justment process for the oxygen can yield accuracies around 1.5 $\mu$mol kg$^{-1}$ (Bittig et al., 2018a), although depending on the details of the optode calibration, handling, and usage scenario, the accuracy of O$_2$ measurements can vary considerably. When O$_2$ sensors incapable of in-air referencing are used (e.g., SBE63 optode, Sea-Bird Electronics), oxygen values typically have uncertainties up to ca. 3% (Takeshita et al., 2013), adding an additional source of uncertainty when these data are used as input parameters to derive reference-pH data.

    In our case, O$_2$ from the 10 pH-equipped Argo floats was adjusted following Argo procedures (Bittig et al., 2018a; Thierry et al., 2022) with in-air measurements and the adjustments are available in near-real time. In February 2023, one float had been recovered and five were still operational. We point out that, unfortunately, the 10 deployed floats suffered from an unusually high number of manufacturer-related technical issues or failures either of the pressure sensor (WMO 3901167, replaced from
125 warranty by WMO 7900566), the GPS system (WMO 7900566) or the pH sensor itself (WMO 6904110, 6904111, 6904112, 6904114, 6904115). This has severely compromised the amount of data acquired so far in the pilot study and reduces the robustness of the conclusions. As the two most long-lasting floats deployed in 2018 (WMO 3901668 and 3901669) showed

stable pH data and the pH sensors have serial numbers not related to a recent problem with the pH sensor's reference electrode, they have been assumed to represent the optimum case for the achievable performance of this current technology. We note,

that the high failure rate points at problems in sensor manufacturing in recent years that need to be resolved in order for BGC-Argo to unfold its full potential. In addition, float-pH data measured by the float WMO 6904112 have been used in this study considering its position regarding the SOOP line corridor and the high number of crossovers recorded. As a consequence, only float-pH data recorded by these three floats are used here. Moreover, adjusted temperature, salinity and oxygen data were available for these floats. Although some Argo float profiles have been reported to be impacted by a "hook" in the oxygen data

at the deepest 50 meters inducing low oxygen values (Wolf et al., 2018), a visual inspection of the oxygen profiles from these three floats did not show this bias.

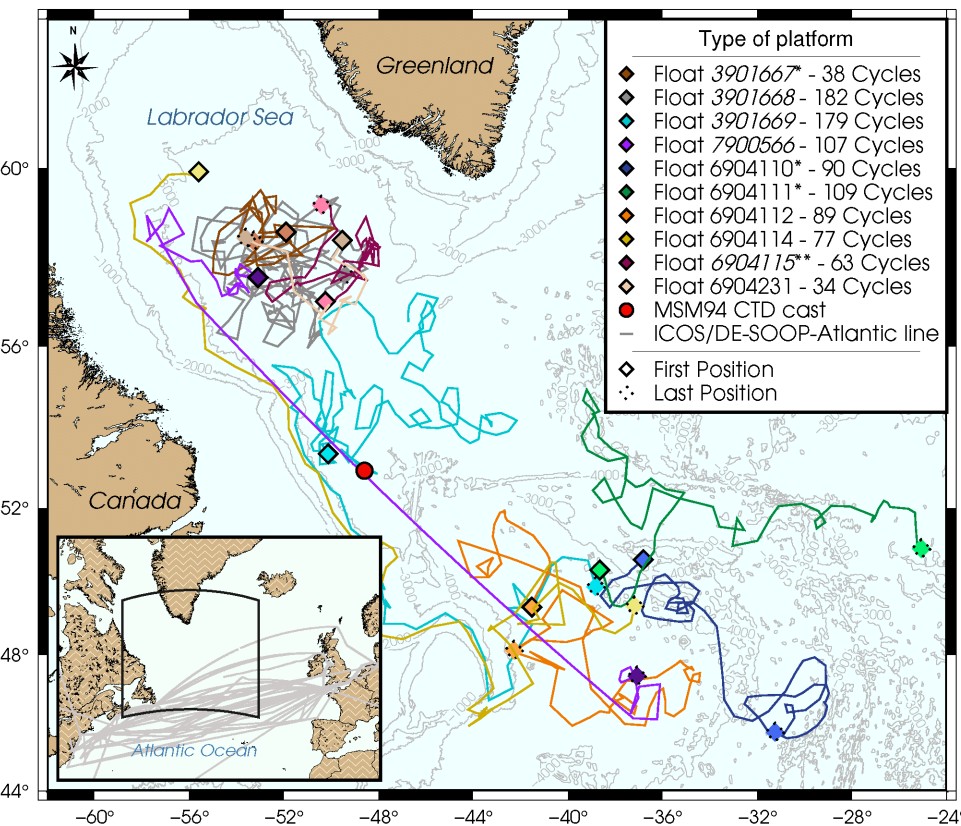

**Figure 1.** Map of the Northwest Atlantic with Labrador Sea and North Atlantic Current showing the trajectories of all 10 pH/O$_2$ floats deployed so far in our pilot study. In the legend, floats in italics are inactive. $^*$ Float with a faulty pressure and/or pH sensor. $^{**}$ Float recovered. Dotted points show the last locations as of February $7^{th}$, 2023. In the inserted map, gray lines indicate the corridor and discrete ship routes occupied by our "Ship-of-Opportunity" platform (ICOS station DE-SOOP-*Atlantic Sail*) during the period 2021-2022. The red dot indicates the location of hydrographic station 13 visited during the *Maria S. Merian* cruise 94 (MSM94) in August 2020

## 2.2 Reference measurements

In situ pH data measured from water samples are generally considered as reference data for float-based observations and are useful tools to independently estimate pH data accuracy and, if needed, apply additional adjustments. Nevertheless, under normal circumstances, it would be nearly impossible to obtain specifically close crossovers between CTD casts and floats profiling during a float's lifetime without significantly impacting the fieldwork schedule of a research cruise. The comparison of discrete pH samples, taken from a hydrocast at the float deployment, with float-pH data is limited due to the high sensor drift during the first cycles (Bittig and Körtzinger, 2015; Bittig et al., 2018a; Maurer et al., 2021). In the Southern Ocean, Maurer et al. (2021) reported an offset value for the first segment of -0.32 pH units, illustrating the sensor performance upon deployment caused by the lack of conditioning in some of the pH sensors as well as the sensor re-conditions to an aqueous environment. However, after float-pH data adjustments, Johnson et al. (2017) and Maurer et al. (2021) showed median shipboard bottle-minus-float differences of 0.006 pH units and 0.002 pH units, respectively. In the SNWA area, we had the unique opportunity to acquire a hydrocast with discrete pH samples with a float profile.

**Table 1.** Crossover between pH profiles of float WMO 3901669 and a CTD cast acquired in the Labrador Sea in August 2020. Time and position refer to the end of the profile.

| Profile | Time | Position |
|---|---|---|
| Float WMO 3901669, Cycle 122 | August 15th, 2020 10:26 UTC | 52.955°N-48.600°W |
| MSM94 CTD cast, Station 13 | August 16th, 2020 05:36 UTC | 52.953°N-48.600°W |

A few float (WMO 3901669) pH profiles occurred close to the R/V *Maria S. Merian* 94 (MSM94) cruise in August 2020 (Karstensen et al., 2020). Thanks to the cooperation of the chief scientist of the cruise and in a joint effort with the Euro-Argo RISE project, a spatio-temporally close crossover was achieved: hydrographic station 13 with discrete sampling for pH analyses was occupied less than 1 day after and at the exact location of the float cycle 122 (Table 1, Fig. 1). The discrete samples were poisoned onboard following standard operating procedures (Dickson et al., 2007). They were measured at GEOMAR for total alkalinity (TA), dissolved inorganic carbon (DIC) and pH. Since DIC and pH are very sensitive to gas exchange they were measured in parallel as soon as the bottles were opened. DIC was measured using a classical SOMMA system (Johnson et al., 1993) with coulometric detection, while pH was measured using the HydroFIA-pH system from 4H-Jena. The pH measurements were checked regularly against community-accepted certified reference material (CRM, Andrew Dickson, Scripps Institution of Oceanography, La Jolla/CA, USA). Note that the CRM is certified only for DIC and TA but pH measurements are also performed routinely for each bag and were made available to us (pers. comm. Andrew Dickson, pH = 7.8417 ± 0.0014 at 25°C). The resulting reproducibility in pH measurements for the discrete samples was ± 0.002 pH units. The pH data were measured at 25°C and atmospheric pressure and were then converted to in situ temperature and pressure using the CO2SYS

software (van Heuven et al., 2011). The matching of float-pH data and discrete pH data was performed in density space rather than depth space to avoid biases from internal wave activity.

In the SNWA, GEOMAR has been operating, with intermissions, a carbon-SOOP line for two decades (ICOS station DE-SOOP-Atlantic Sail; Fig. 1). This SOOP network can be used as a potential reference for quality control of autonomous platform datasets. In addition to the standard $pCO_2$ instrument (Model 8050 $pCO_2$ Measuring System, General Oceanics, Miami/FL, USA; Pierrot et al., 2009), autonomous systems for TA (Contros HydroFIA$^{TM}$ TA system, 4H-JENA engineering GmbH, Jena, Germany) and pH measurements (Contros HydroFIA$^{TM}$ pH system, 4H-JENA engineering GmbH, Jena, Germany) were installed on this SOOP line in 2019 and 2021, respectively. For pH, pre- and post-calibration runs against CRM from Andrew Dickson's laboratory are performed before and after each 5-week roundtrip and an individual pH correction is applied to each pH indicator bag (meta-cresol purple; MCP). The overall reproducibility of SOOP-pH is estimated to be about $\pm$ 0.003 pH units.

## 2.3 Correction of float-pH data

Conceptually, the pH correction has to be done by adjusting the sensor's reference potential ($k_0$) as this is drifting over time (Johnson et al., 2016). For each pH sensor, the in situ pH is proportional to the voltage between the ion sensitive field effect transistor (ISFET) source and the reference electrode (Johnson et al., 2016). The measured potential is then converted into pH on the total proton scale using laboratory-based calibration coefficients. Thus, pH sensors are calibrated in the laboratory using spectrophotometric measurements and are therefore directly related to the laboratory calibration method. Each sensor's pressure and temperature coefficients, needed to compute the in situ pH, are also determined in the laboratory as described in Johnson et al. (2016). When deployed at sea, temperature changes modify the reference potential of the sensor and in return induce a sensor drift as the Nernst slope that transforms sensor potential to pH depends on temperature (Johnson et al., 2016, 2017).

The general adjustment process performed in the SAGE procedure is based on the assumption that the determined offsets and drifts are constant across the entire water column profile (Johnson et al., 2017; Maurer et al., 2021). Thus, the standard SAGE adjustment process relies on a reference that is used to calculate the at-depth (typically around 1500 dbar) anomaly between measured and estimated reference data, which is applied as an offset to the reference potential. It is propagated on the entire water column profile by normalizing the adjustment along the profile to the temperature at which the adjustment was derived. Temperature-normalized changes in pH are calculated by multiplying the change in pH computed at depth by the ratio of the absolute temperature of the sample to the absolute temperature at reference depth. To calculate the correction, the float-pH time-series is split into distinct segments bound on either side by breakpoint nodes determined by a cost function. Then, both drift and offset between segments are calculated by linear least-squares fit to the anomaly data series between two nodes. Indeed, by breaking the time-series sensor record into different segments and fitting each with a linear rate of change in $k_0$, the adjustment better represents the sensor behavior over time as both drifts and offsets change independently between segments and oftentimes noticeable jumps occur over the first few cycles in a float's life (Maurer et al., 2021).

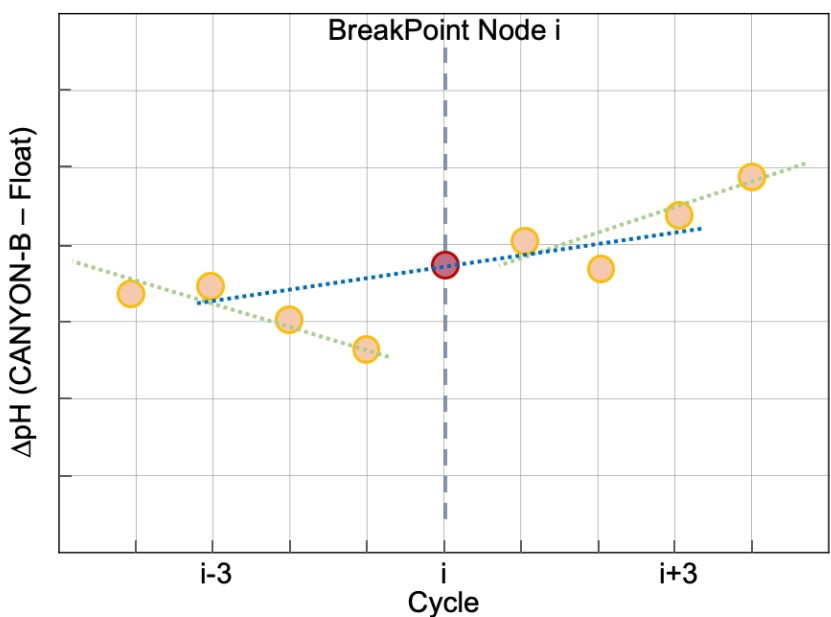

**Figure 2.** Schematic representation of the GEOMAR float-pH data adjustment method called 'linear adjustment'. As in the SAGE tool, a linear least-square fit is calculated between reference and float-pH data for cycles located between two breakpoint nodes to derive the offset and drift (green lines). The blue line represents the second least-square fit obtained and applied to the elements located 3 cycles before and after the node (red dot) in the 'linear adjustment' method. Adapted from Maurer et al. (2021).

In our analysis, three pH correction methods called 'cycle-by-cycle', 'linear adjustment' and '3-point running mean', respectively, have been implemented locally. Like in the SAGE tool, the pH adjustment is calculated by these methods based on comparison to CANYON-B reference pH values calculated at a user-defined pressure level, where spatiotemporal variability of oceanic components is assumed to be minimal. The CANYON-B method was chosen as a reference assuming it to be more robust in the North Atlantic region (Carter et al., 2021). Nonetheless, two slight differences exist between SAGE and the methods

proposed here: (1) The adjustment can be applied either to each cycle individually ('cycle-by-cycle' method) or, as in SAGE, to data within segments of consecutive profiles ('segment method', with each segment calculated using a cost function). (2) When using the segment method, a centered 7-point linear regression is used for cycles neighboring segment breakpoints to allow for a smoother $k_0$ drift between segments (Fig. 2; Johnson et al., 2016). As in SAGE, offset and drift calculated with this method ('linear adjustment' method) are then applied to the measured float-pH profiles after normalization to the temperature

at which the adjustment was derived. Finally, another adjustment method entitled '3-point running mean method' was tested in this study. In this, the adjustment calculated by the 'cycle-by-cycle method' was used to determine a new offset for each cycle calculated as a running mean of 3 cycles, i.e., including the cycle before and after the respective cycle. This method should smooth the adjustment obtained with the cycle-by-cycle adjustment. Hereafter, every adjustment method different from the one in SAGE will be labelled as 'GEOMAR method'.

## 2.4 Comparisons with SOOP-based observations

To compare SOOP-based and float-based surface pH observations, we adopted the crossover definition from the Surface Ocean $CO_2$ Atlas (SOCAT; Sabine et al., 2013) which combines the mismatch in both distance and time between two measurements. In the SOCAT algorithm, one day of separation in time (t in days) is heuristically equivalent to 30 km of separation in space (x in km) and 80 km is the maximum value for an acceptable single crossover $((dx^2+(dt*30)^2)^{1/2}$; Wanninkhof et al., 2013). Here we used a much increased search window of 400 km to yield a larger number of crossovers and to optimize between spatial and temporal mismatch. In addition, a maximum temporal mismatch of 7 days was allowed for a crossover. The SOCAT criterion of a maximum of 80 km aims to compare two datasets of surface $pCO_2$ observations to agree better than 2 $\mu$atm. In this study, we conclude that this is not yet routinely achieved by pH data from floats and therefore we used a larger radius to ensure more crossovers and better statistics. The resulting crossovers were further reduced by the requirement of a maximal salinity difference between the float measurement and the salinity measurement onboard the SOOP line of 0.5 $(-0.5 < \Delta S < 0.5)$. To make the pH measurements from both platforms comparable, the SOOP-based pH data were corrected to the surface water temperature of the corresponding float profile. We note that for possible future implementation of the SOOP crossover method in the DM QC routine for float-pH data this needs to be further explored and more elaborate crossover criteria may have to be developed.

## 2.5 MLD calculations

Following De Boyer Montégut et al. (2004), a density threshold of 0.03 kg m$^{-3}$ with a reference depth of 10 dbar was used to compute the Mixed Layer Depth (MLD). We used MLD to determine waters affected by deep convection events which cause unstable biogeochemical properties also at depth that are being used for float-pH data adjustments.

## 3 Results and Discussion

### 3.1 Uncertainties of delayed-mode float-pH data

In the following we first illustrate uncertainties associated with the current correction method for float-pH data as implemented in the standardized routines from Argo data management as well as in the SAGE tool for four referencing methods (CANYON-B, ESPER-NN, ESPER-LIR and LIR-pH) and two selected floats (WMO 3901668 and 3901669) which had no apparent technical malfunctions during their lifetime.

#### 3.1.1 Uncertainty associated with choice of reference depth

In order to assess the uncertainty associated with the choice of the reference depth for pH adjustment, differences between float-pH data corrected using the "classical" reference pressure around 1500 dbar (Maurer et al., 2021) minus float-pH data corrected over the pressure range 1940-1980 dbar (i.e., pressure around 1950 dbar) were calculated for the four reference

methods LIR-pH (without the OA adjustment), CANYON-B (Fig. 3A), ESPER-NN and ESPER-LIR (Fig. 3B).

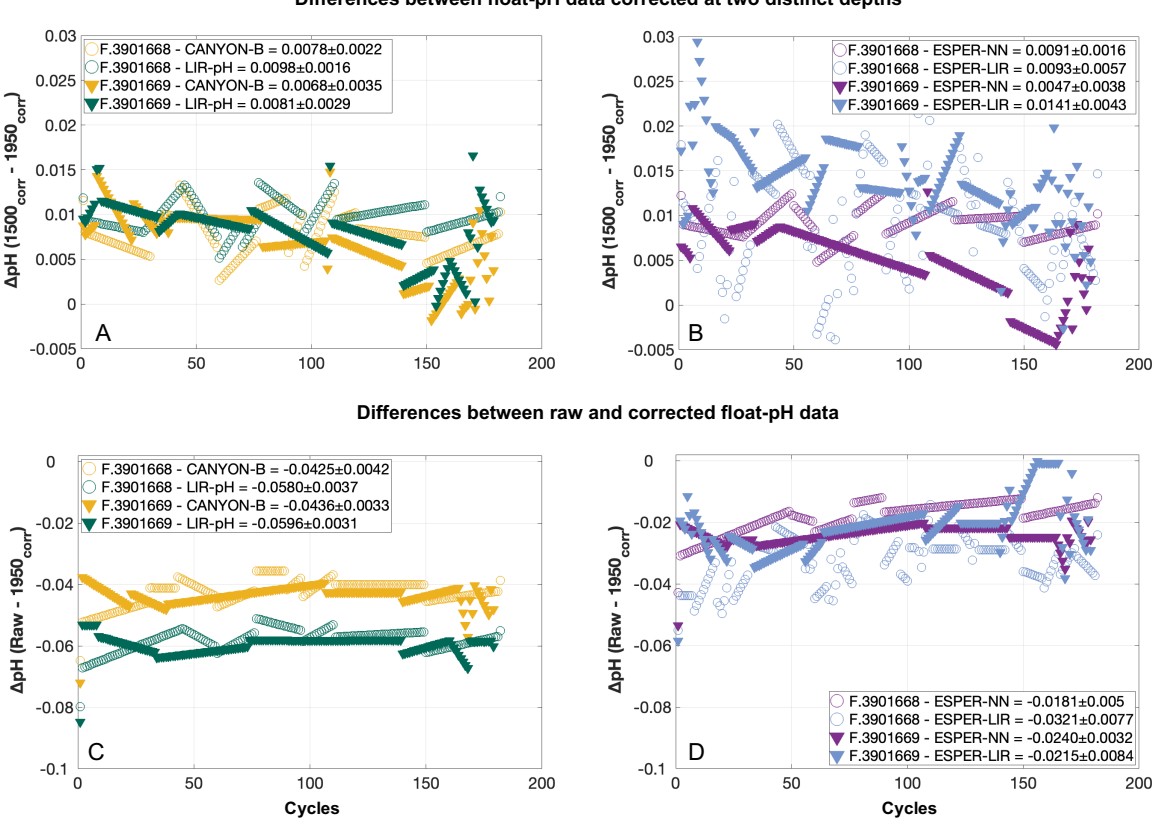

**Figure 3.** (A and B) Mean differences between float-pH data corrected using the "classical" reference depth of 1500 dbar minus float-pH data corrected with reference-pH data calculated between 1940 and 1980 dbar (i.e., 1950 dbar) for the floats WMO 3901668 (circle dots) and 3901669 (triangle dots) and for the reference methods (A) LIR-pH without the OA adjustment (green dots) and CANYON-B (orange dots) and (B) ESPER-NN (purple dots) and ESPER-LIR (blue dots). (C and D) Raw float-pH data minus float-pH corrected using the "area-specific" reference depth of 1950 dbar for the two-reference methods CANYON-B (orange dots) and LIR-pH (green dots) in Fig. 3C and the two-reference methods ESPER-NN (purple dots) and ESPER-LIR (blue dots) in Fig. 3D and for the floats WMO 3901668 (circle dots) and 3901669 (triangle dots).

Differences between float-based pH data for the two difference reference depths as achieved by the four methods ranged between -0.0005 and ca. 0.03 pH units, with mean values for all cycles of the considered floats varying between 0.0047 and 0.0141 pH units (Fig. 3B). The choice of the reference depth thus incurs a large difference of at least ca. 0.005 pH units which is above a tolerable level. This points to a severe limitation of the pH correction scheme. The deepest mixed layer

depth estimated from the float time-series was at 1937 dbar, showing that the entire water column covered by the float profiles is probably affected. In this regard, the subpolar North Atlantic region with its deep-reaching anthropogenic $CO_2$ imprint is

certainly a most difficult area for the unambiguous choice of a stable and unperturbed reference depth as both float-pH data and reference pH values could vary noticeably at the classical reference depth. By splitting the dataset to keep only profiles done when the MLD was deeper than 1000 dbar, the comparison between raw and corrected float-pH data using the two reference pressures reveals larger variabilities when the classical reference depth of 1500 dbar is used as compared to the deepest one, highlighting the implication of deep convection events on the adjustment method (Table A1). Recently, Wimart-Rousseau et al. (2022) performed a similar exercise by changing the reference depth from ca. 1500 dbar to ca. 900 dbar for a float in the Eastern Tropical North Atlantic region and reported a tolerable uncertainty from this choice of 0.0008 pH units. The order of magnitude difference in the uncertainty incurred from the reference depth choice illustrates the regional dependence on hydrological conditions which can severely compromise the correction method or even render it almost useless as in the case presented here.

### 3.1.2 Uncertainty associated with choice of reference model

Four distinct reference methods are used in the standardized Argo pH quality control, both in SAGE and in this study: the LIR pH regression method (LIR-pH), the CANYON-B method (Fig. 3C), the ESPER-NN and the ESPER-LIR mehods (Fig. 3D). For all methods, corrected float-pH showed significant mean offsets to the raw pH profiles comprising values between ca. -0.02 and -0.06 pH units (Fig. 3C and D). Moreover, mean differences between the four reference methods ranging between about 0.003 pH units and ca. 0.04 pH units are observed in the SNWA, with the lowest difference reported for the ESPERs methods indicating that they perform comparably (Table 2).

While the CANYON-B and the LIR-pH algorithmic methods are methodologically different (one is based on a neural network while the other uses linear regressions), both have been trained with and tested against the GLODAPv2 dataset (Olsen et al., 2016). Still, ocean pH measurement practices have changed over time, leading to a variety of ways to measure pH. In addition, pH calculated from DIC and TA is not always in line with spectrophotometrically measured pH (Carter et al., 2018). In consequence, heterogeneities in pH data compilations such as GLODAPv2 exist. While CANYON-B was trained with GLODAPv2 without modifications, Carter et al. (2018) applied a range of adjustments to create a more consistent pH data product that was used for LIR-pH training (with pH being in line with "purified spectrophotometric pH measurements"). Given the dominance of calculated pH data in GLODAPv2, CANYON-B pH estimates are in line with calculated pH (Bittig et al., 2018b; Carter et al., 2018). In the SAGE software, an optional CANYON-B pH data adjustment can be applied to align estimates with spectrophotometric pH measurements made using purified dye following Carter et al. (2018, Equation 1). The recent literature (Carter et al., 2018; Johnson et al., 2018) recommends employing this reference-pH data adjustment emphasizing that, as pH sensors are calibrated in the laboratory using spectrophotometric measurements with purified dyes, sensor measurements should be directly related to the laboratory calibration method. In this study, we have decided to include this reference-pH data adjustment to correct float-pH data: a linear transformation was applied to CANYON-B pH estimates to bring estimates back into alignment with spectrophotometrically measured pH. For the two floats considered in this section, means and standard deviations of the difference between float-pH data corrected at 1500 dbar using CANYON-B and CANYON-B adjusted are equal to $0.0055 \pm 6.63 \times 10^{-5}$ and $0.0055 \pm 8.31 \times 10^{-5}$, respectively. The ESPERs routines

broadly function similar to LIR and CANYON-B although using a gridded anthropogenic carbon product to estimate the OA, assuming a marine anthropogenic carbon increase proportional to an exponential increase in atmospheric anthropogenic $CO_2$ concentration. Therefore, it is more critical than ever for the scientific community to perform intercomparisons of marine $CO_2$ system variables and address their associated uncertainties regarding the large and growing variety of instruments and

285 approaches used to measure, deduce and calculate $CO_2$ variables. Fig. 4 exhibits spatial distributions of estimated pH data at the classical reference 1500 m depth level using either LIR-pH (with the OA adjustment), CANYON-B, ESPER-LIR, or ESPER-NN and illustrates the differences between the estimated datasets with uncertainty between reference algorithms in the order of 0.015 pH units in the SNWA area. Despite the undeniable strength of current algorithms, CANYON-B and LIR-pH methods suffer from weaknesses and uncertainties due to the pH adjustment: a complete regional or temporal description of the

290 current ocean acidification is limited with LIR-pH (i.e., LIR-pH assumes fixed OA rates over time; Carter et al., 2018), and the pH conversion according to another measurement mode in CANYON-B induces biases (Bittig et al., 2018b). In consequence, a mean difference between the two methods of about $\pm$ 0.016 pH units is observed in the SNWA (Table 2 and Fig. 4).

**Table 2.** Mean differences (y-x) between float-pH data corrected at two distinct depths and using the four different reference methods for the floats WMO 3901668 and 3901669. SD stands for Standard Deviation.

| | y-x | 1500 db | | | | 1950 db | | | |
|---|---|---|---|---|---|---|---|---|---|
| | | LIR-pH | CANYON-B | ESPER-LIR | ESPER-NN | LIR-pH | CANYON-B | ESPER-LIR | ESPER-NN |
| WMO 3901668 | LIR-pH | / | $0.0175 \pm 0.0012$ | $0.0264 \pm 0.0026$ | $0.0407 \pm 0.0028$ | / | $0.0155 \pm 0.0011$ | $0.0259 \pm 0.0052$ | $0.0399 \pm 00027$ |
| | CANYON-B | $-0.0175 \pm 0.0012$ | / | $0.0089 \pm 0.0016$ | $0.0232 \pm 0.0019$ | $-0.0155 \pm 0.0011$ | / | $0.0105 \pm 0.0045$ | $0.0245 \pm 0.0024$ |
| | ESPER-LIR | $-0.0264 \pm 0.0026$ | $-0.0089 \pm 0.0016$ | / | $0.0143 \pm 0.0013$ | $-0.0259 \pm 0.0052$ | $-0.0105 \pm 0.0045$ | / | $0.0140 \pm 0.0053$ |
| | ESPER-NN | $-0.0407 \pm 0.0028$ | $-0.0232 \pm 0.00196$ | $-0.0143 \pm 0.0013$ | / | $-0.0399 \pm 0.0027$ | $-0.0245 \pm 0.0024$ | $-0.0140 \pm 0.0053$ | / |
| WMO 3901669 | LIR-pH | / | $0.0173 \pm 0.0014$ | $0.0321 \pm 0.0069$ | $0.0391 \pm 0.0021$ | / | $0.0161 \pm 0.0017$ | $0.0381 \pm 0.0076$ | $0.0356 \pm 0.0018$ |
| | CANYON-B | $-0.0173 \pm 0.0014$ | / | $0.0148 \pm 0.0058$ | $0.0217 \pm 0.0014$ | $-0.0161 \pm 0.0017$ | / | $0.0221 \pm 0.0070$ | $0.0196 \pm 0.0014$ |
| | ESPER-LIR | $-0.0321 \pm 0.0069$ | $-0.0148 \pm 0.0058$ | / | $0.0069 \pm 0.0055$ | $-0.0381 \pm 0.0076$ | $-0.0221 \pm 0.0070$ | / | $-0.0025 \pm 0.0074$ |
| | ESPER-NN | $-0.0391 \pm 0.0021$ | $-0.0216 \pm 0.0021$ | $-0.0069 \pm 0.0055$ | / | $-0.0356 \pm 0.0018$ | $-0.0196 \pm 0.0014$ | $0.0025 \pm 0.0074$ | / |

In addition, using the SOCCOM array, Maurer et al. (2021) calculated LIR-pH and CANYON-B pH estimates and observed a larger uncertainty toward the surface compared to 1500 m with mean differences (CANYON-B minus LIR-pH pH data) of -

295 0.025 and 0.001 pH units near the surface and at the 1500 m depth level, respectively. This surface discrepancy can be explained by the difficulty for algorithms to represent seasonal variability and air-sea gas exchange. The new ESPERs methods attempt to resolve the issues encountered with existing routines (especially the OA estimate) by expanding their functionality and being trained on a larger data product. In comparison with the LIR-pH estimates, large differences are observed in the SNWA region and might be attributable to the OA adjustment as well as the omission of depth as a predictor variable from ESPER-LIR

(Carter et al., 2021). Updated global algorithms (i.e., ESPERs) show comparable estimates in the SNWA area with ESPER-LIR pH estimates slightly higher than pH data estimated with CANYON-B or ESPER-NN. In the dynamic and strongly human-impacted studied region, the lack of coordinate information as a predictor variable in the ESPER-LIR routine could also be argued as an explanation of the observed differences. However, according to Carter et al. (2021), regional assessment statistics obtained in the Northern Atlantic indicate almost similar biases for both the ESPERs and the CANYON-B methods, with a

better RMSE statistic for CANYON-B. Thus, this study illustrates the need for further studies on the choice and performance of the referencing method in different ocean regions with a special emphasis on regional biases and limitations.

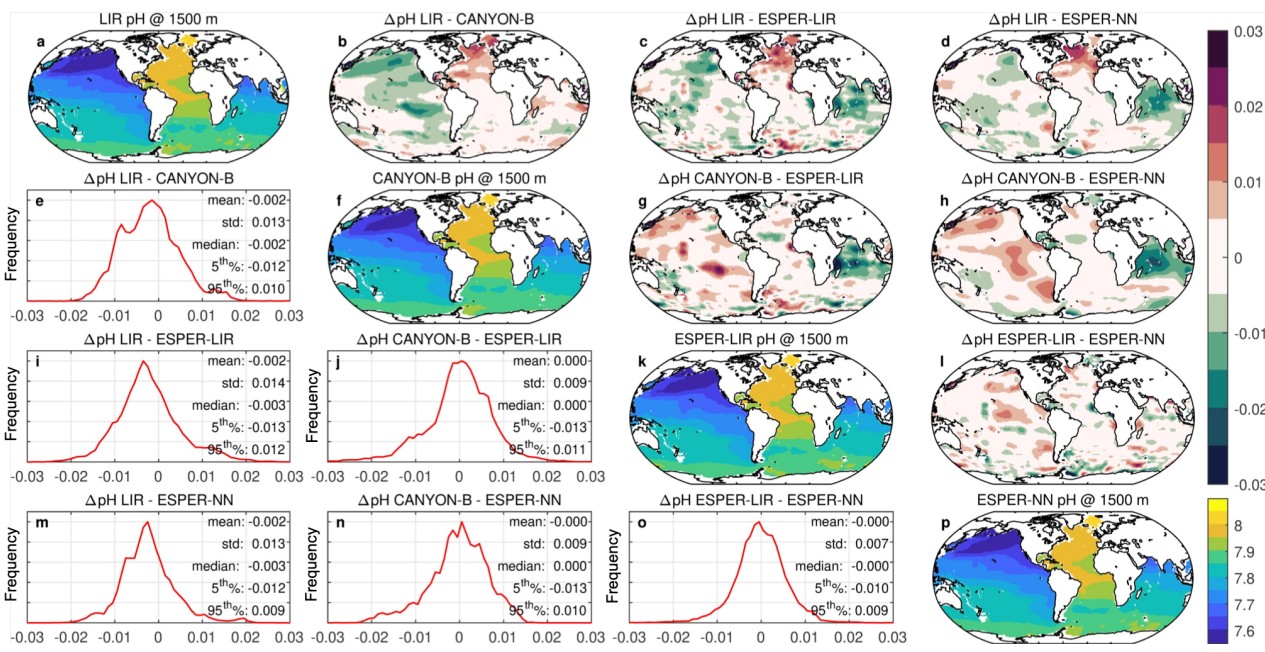

**Figure 4.** Spatial distributions of estimated pH data at the classical reference depth 1500 m using different reference models: LIR-pH (with the OA adjustment) (a), CANYON-B (f), ESPER-LIR (k), and ESPER-NN (p). Maps of the spatial difference between the estimated pH datasets are presented in panels (b-d, g-h and i). Panels (e, i-j and m-o) show the bias ΔpH distribution (with statistics). The upper colorbar indicates the difference between estimated pH data using the different models and the lower colorbar gives the pH values. For clarity, pH data estimated for the Black Sea, the Baffin Bay, the Mediterranean Sea and the High Arctic have been removed for this simulation as they were outside the $5^{th}/95^{th}$ percentile and they caused a noticeable increase of the standard deviation (std). World Ocean Atlas climatology data was used to create the maps and comparisons.

### 3.1.3 Adjustment of sensor drift

In addition to the choice of reference depth and method, some additional uncertainty can be incurred from the way the pH sensor drift correction is applied to the float data. Sensor response often shows different modes of variability and drift. A typical 310 mode of variability is sensor noise, i.e., variability entirely introduced by electronic components of the sensor. This noise does not represent true variability in the observed quantity and should therefore be removed. In addition, long-term systematic drift in sensor response due to changes in zero levels and/or gain factors is also an internal artefact of the sensor that needs to be corrected for. More rarely, sensors can also show more erratic and non-systematic variability in individual measurements or over certain measurement periods which often has unknown reasons. These are hard to distinguish from true variability in the 315 observed quantity and are hence also hard to be removed. The method to apply sensor corrections in time-series measurements

should take a conservative approach trying to remove known modes of sensor variability while conserving real variability in the data.

In the sequence of steps in the current delayed mode Argo adjustment method, first the $\Delta$pH (raw- corrected, at reference depth) is calculated for each cycle. In the SAGE tool, a cost function is applied for the correction of temporal trends which determines sections over which a linear correction is calculated and then applied to each cycle included in the respective section (Fig. 5A). We also applied three different adjustment methods: (1) a cycle-by-cycle method, (2) a 7-point linear regression method named "linear adjustment" and (3) a 3-point running mean adjustment method, which should smooth the correction obtained with the cycle-by-cycle adjustment (Fig. 5B). In every case, CANYON-B was used as the reference method as well as the "classical" reference pressure depth of 1500 dbar.

The choice of the correction method has to reflect our understanding of the sensor's behavior. Over time, sensor reference potential shifts are observed for pH sensors, leading to jumps in the data time-series. As stated by Maurer et al. (2021), these jumps are typically periodic and followed by longer periods of steady drift. The cycle-by-cycle adjustment has the disadvantage that it gives discontinuous adjustment rather than a segmented set of piecewise adjustments. On the other hand, a single linear drift adjustment across the entire time-series does not seem adequate either as it does not reflect the clear upward and downward swings in the record which are mostly interpreted as changes occurring in the sensor. Therefore, the adjustment method should involve techniques such as a higher-order spline fit, a centered running mean, or a segment separation of the record into linear drift phases. The latter is implemented in the SAGE tool (Fig. 5A). This method, however, does not provide smooth transitions between linear drift phases and leads to step-like changes of the order of 0.01 pH units between two consecutive profiles which appear to be unrealistic when compared to the pattern of the cycle-by-cycle correction and the pH readings at the parking depth. The correction methods for temperature and salinity also ask for maximum smoothness in the corrections and to avoid introducing artificial jumps (Owens and Wong, 2009). Our slightly improved GEOMAR linear adjustment version (Fig. 5B) significantly reduces the magnitude of these discontinuities and artificial jumps. Generally, the linear segment methods assume periods of linear sensor drift separated by step-like changes in sensor characteristics. In our view, the sensor rather shows undulations between smooth and less smooth phases. The pH sensor behavior when the float drifts at its parking depth is in agreement with this observation (Fig. 5C). In comparison with float-pH data corrected using the SAGE method, no obvious discontinuities in raw pH data are observed while the float drifts between its measurement phases as well as on the uncorrected float-pH time-series measured at 1500 dbar (Fig. 5E and F). In order to test the impact of the reference method on the adjustment pattern, differences between uncorrected float-pH data and CANYON-B pH data derived at the parking depth are presented in Fig. 5D. Moreover, high variability is observed on the reference pH time-series estimated using both CANYON-B (Fig. 5E) or ESPER-LIR (Fig. 5F), highlighting the noticeable impact of the reference algorithms discontinuities on the final correction while raw float-pH data are not presenting sudden changes. Indeed, the raw pH time-series shows smoothed transitions and the general pattern does not present noteworthy jumps. Such sharp transitions can perhaps be best corrected with our modified GEOMAR segment method or alternatively with a spline fit or a 3-point centered running mean (Fig. 5B).

We suggest to use the improved segment or running mean method to avoid strong discontinuities in the pH correction which otherwise could introduce biases in corrected pH of up to 0.01 pH units in individual profiles – a magnitude that would

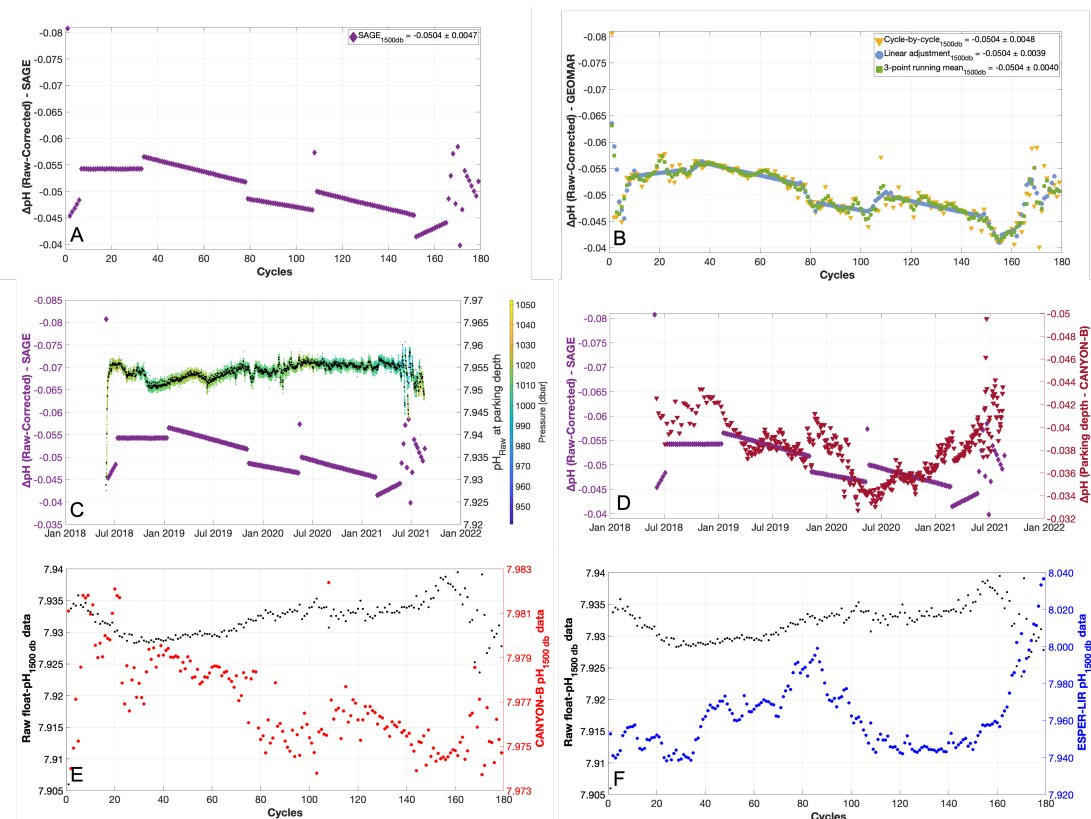

**Figure 5.** Differences between raw float-pH data minus float-pH corrected using the SAGE tool (Fig. 5A), the cycle-by-cycle GEOMAR method (yellow dots, Fig. 5B), the linear mean regression GEOMAR method (blue dots, Fig. 5B), and the 3-point centered running mean correction method (green dots, Fig. 5B) for float WMO 3901669. In every case, CANYON-B was chosen as a reference method, and 1500 dbar as the reference depth. Mean differences between raw and corrected float-pH data with the standard deviations are shown in the legend boxes for each reference method. Fig. 5C shows, for comparison with the SAGE correction, the uncorrected pH data measured at the parking depth (right y-axis) with black dots representing mean pH values for each day. The color bar shows pressure. Fig. 5D shows differences between raw float-pH data minus float-pH corrected using the SAGE tool (purple dots, left y-axis) and differences between uncorrected mean pH data measured at the parking depth minus mean reference CANYON-B pH data calculated using measurements recorded at the parking depth (red dots, right y-axis). Figs. 5E and 5F show mean raw float-pH data measured around 1500 dbar (between 1480 and 1520 dbar) and pH data calculated by the reference methods CANYON-B (panel 5E) and ESPER-LIR (panel 5F) using as input parameters (i.e. temperature, salinity, pressure and oxygen) the values measured by the float at 1500 dbar. For panels A to D, differences are calculated for each cycle at each depth along the entire profile and then averaged.

strongly impair quality control measures based on referencing against other in situ pH measurement from CTD casts or surface observation platforms (see Section 3.2). Indeed, and even if the impact of the adjustment method on the final corrected dataset is almost non-significant regarding the mean difference values (Fig. 5D), the possible impact of such artificial jumps induced

by the method itself rather than the pH sensor could be noticeable if float-pH data related to these peculiar discontinuous cycles are compared against discrete pH measurements and then adjusted (see Section 3.2).

## 3.2 Comparison with in situ discrete pH

### 3.2.1 Crossover with CTD hydrocast

Crossover comparisons can be used as an option to independently estimate float-pH data accuracy and determine if additional adjustments are needed. In 2020, we had the rare opportunity to perform a CTD hydrocast with discrete pH sampling (cruise MSM94) at the exact location and less than 24 h after a float profile (WMO 3901669, profile 122; Fig. 1) which allows for direct comparison between discrete and float-based in situ pH data after the float's initial drift period. Figs. 6A and 6B present differences between discrete pH measurements and float-pH data along the water column and according to two distinct reference pressure levels. We find mean differences ranging between -0.0659 and -0.0150 pH units (Fig. 6B) between the reference pH cast and the fully corrected pH of cycle 122, with higher differences found for the "classical" reference depth of 1500 dbar (Fig. 6A), and the lowest differences reported for the two ESPER methods.

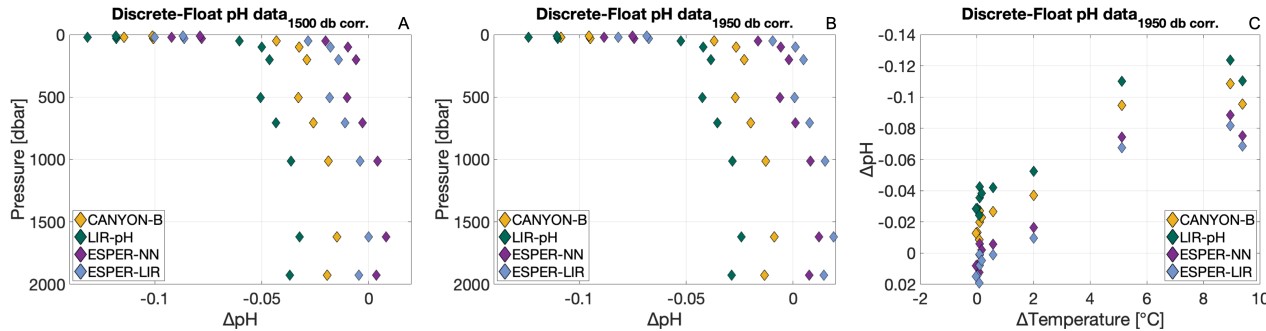

**Figure 6.** (A and B) Differences between discrete and float-pH data (for the cycle 122) calculated after matching in density space to avoid biases from internal waves and corrected using corrected reference levels of 1500 dbar (Fig. 6A) and 1950 dbar (Fig. 6B). (C) $\Delta$pH (discrete pH measurements minus float-pH data corrected at the reference depth level 1950 dbar) as a function of the difference between discrete water temperature (i.e., the temperature measured in situ at the time of bottle triggering at sea) and temperature values recorded at the reference depth of 1950 dbar. The color code refers to the reference method used to correct float-pH data: CANYON-B (yellow diamonds), LIR-pH (green diamonds), ESPER-NN (purple diamonds) or ESPER-LIR (blue diamonds).

Matching sensor data from a float with discrete samples is a non-trivial task due to complications arising from (a) the sensor response time and (b) the uncertainty about the effective depth from which the water captured in a Niskin bottle at a trigger given depth stems from. There seems to be no perfect way of matching these and some uncertainty remains – especially in-depth ranges with strong gradients in the variable of interest. Mismatch (and resulting statistical noise) due to internal wave activity can mostly be avoided by matching profile and bottle data in density space which was performed here. Differences between discrete and float temperature and salinity data add confidence in the density space matching performed in this study

(Fig. A1). However, the likely imperfect representation of the true water sampling depth by the trigger depth (and hence corresponding CTD data) of a Niskin bottle introduces the potential of systematic error in gradient regimes, although in a gradient of increasing pH both effects (a) and (b) would lead to underestimation of pH. Still, the results of this comparison therefore have to be interpreted with caution. Moreover, the laboratory-to-in-situ temperature pH conversion uncertainty of 0.005 pH units (Williams et al., 2017), as well as the absolute reproduciblity in the bottle pH measurements (here 0.002 pH units), have to be taken into account before drawing strong conclusions.

The results show the smallest offsets at/near the reference pressure levels and an increase towards the surface. In this area, near-surface variability and patchiness can be large and would require a perfect match in both space and time for strong conclusions and robust significance of the surface values observations (< 30 dbar). Nevertheless, pH offsets are positively correlated with temperature, being smallest at the temperature of the reference depth. Overall, the results appear to be robust and not an artefact of the matching procedure and point towards an imperfect representation of the temperature and pressure dependences of the pH sensor (Fig. 6C). Although the actual pH values may be slightly different due to the regional variability, the observed trend is confirmed. However, this single crossover does not allow for a solid conclusion and therefore can only serve as a hint at shortcomings in the pH referencing method. With larger amounts of matchups between hydrocasts and pH profiles and optimized SOOP-float crossover data, an independent validation and perhaps adjustment method could be investigated. Indeed, SOOP data can represent an additional reference and comparison data source.

### 3.2.2 Crossover with SOOP-based surface measurements

In addition to the comparison of entire pH profiles as described above, we compared float-based pH measurements in the surface (average pH between 5 and 15 m depth) with surface pH measurements from a SOOP line crossing the North Atlantic twice every 5 weeks (see Section 2.4). The cruise track of the SOOP line partly overlaps with the trajectories our floats deployed in the North Atlantic region (see Fig. 1). For this comparison, we used data from two floats (WMO 6904112 and WMO 3901669) between May 2021 and October 2022. We note that further testing and improvement of this approach on larger datasets needs to be carried out to define an optimal crossover criterion. Given the limitations of the dataset (mostly due to massive manufacturing problems of the 2020/2021 pH sensor series), unfortunately no robust recommendations in terms of absolute numbers can be drawn from these experiments. Nevertheless, the assumption was made hereafter that regressions using crossovers achieved with a relatively wide search window yield a more robust $\Delta$pH estimate than an average of a small number of crossovers found with a smaller search window.

Fig. 7 shows the differences ($\Delta$pH = SOOP - float) between SOOP-based surface pH observations (corrected to the temperature of the respective float surface pH observations) and the averaged mixed layer pH values of the two pH/$O_2$ floats as a function of $\Delta T$ (temperature difference between float data and SOOP data). While we found no dependence between $\Delta$pH and $\Delta S$, an additional criterion of $\Delta S \leq 0.5$ has been applied to the crossovers selection in order to exclude major water mass discrepancies. As any mismatch in temperature will likely be associated with a corresponding mismatch in pH (both due to the temperature sensitivity of pH and different water mass properties), the $\Delta$pH at $\Delta T = 0$ should be a reasonable estimate of the pH offset between SOOP and float. By fitting a linear regression to the data, the pH offset at $\Delta T = 0$ can be estimated more

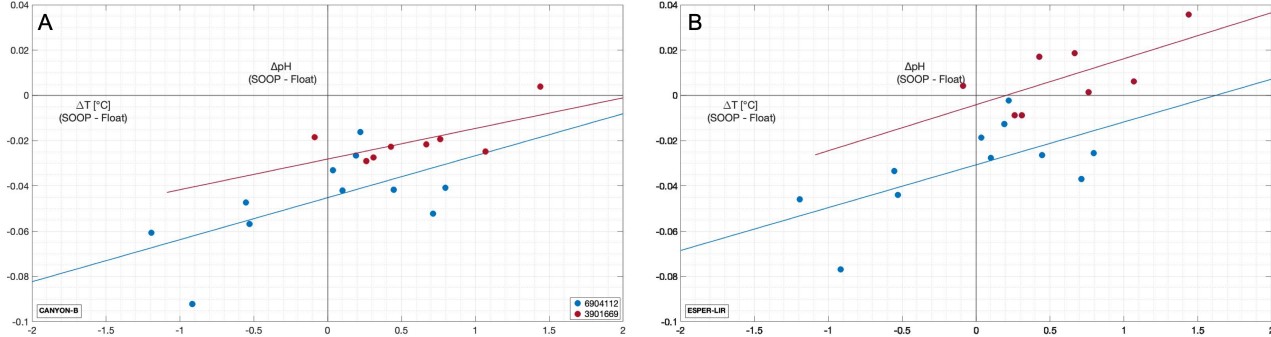

**Figure 7.** Offsets between SOOP pH and fully corrected float-pH (y-axis) as a function of temperature difference (x-axis) for crossovers ($\Delta x \leq 400$ km, $\Delta t \leq 7$ d, $\Delta S \leq 0.5$) of two different floats. Float-pH data have been corrected with the SAGE tool using the reference depth level 1950 dbar and either CANYON-B (left panel) or ESPER-LIR (right panel) as reference.

robustly as the intercept of the regression equation. We want to point out that this analysis has its limitations: (1) the study area is characterized by high spatiotemporal surface variability due to mixing of water masses of very different provenience, (2) the presented analysis uses only data from 2 floats during an 18-month period. However, the comparison between float-based pH and SOOP-based pH data indicates that surface pH is very consistently biased high for the two floats (between ca. 0.05 and

410 ca. 0.004 pH units depending on the choice of correction methods). This apparent bias is in the same direction (albeit about a factor of 3 smaller) than what was found in the comparison with discrete CTD cast samples for surface waters. This suggests a systematic problem with float-based pH measurements in the surface.

The average $\Delta T$ of the crossovers for the two floats is of 0.22°C corresponding to a mean $\Delta S$ of -0.003 going in the opposite direction. This indicates that the crossovers identified for each float are a reasonable but not perfect match. Calculating the

415 apparent pH offset as a function of $\Delta S$ (Table and Fig. A2) yields $\Delta$pH values which are statistically indistinguishable from the ones based on $\Delta T$. Table 3 shows the pH offsets and their uncertainties for two floats and two pH correction methods as given by the intercept ($\pm$ uncertainty) of the linear regression fit to the data. We note that the error associated with the pH offset is too large to be applied as correction. However, despite the limited number of floats and crossovers associated with this study, the preliminary results point at unacceptably high and almost identical biases in surface pH values from the 2 floats (as seen by

420 the values crossing the y-axis), which have been corrected in the exact same way. An extended crossover comparison with the addition of four floats (that were not part of our pilot study) yields mean pH offsets that fall in the range $\pm$ 0.03 pH units (Fig. A2). These mean pH differences are randomly distributed in space and time (Figs. A2), indicating an incomplete float-pH data adjustment rather than a drift in the SOOP-reference dataset. This highlights that the present instructions to correct pH with a unique offset established at-depth are insufficient, at least in our study area. An improved understanding of the temperature

(and pressure) effect on the (individual) sensor as well as a systematic adjustment with carbon measurements could be the way forward to improve float-pH data adjustment.

**Table 3.** Statistics of the crossover analysis for SOOP- and float-pH data.

| | | $\Delta$pH at $\Delta T=0$ | |
|---|---|---|---|
| | *Float WMO* | **pH offset** | **Uncertainty of the offset** |
| **CANYON-B** | 3901669 | -0.029 | 0.012 |
| | 6904112 | -0.047 | 0.014 |
| **ESPER-LIR** | 3901669 | -0.004 | 0.007 |
| | 6904112 | -0.031 | 0.014 |

## 3.3 Implications and changes in ocean chemistry

BGC-Argo float-based pH data can potentially be a very powerful tool to estimate the ocean $CO_2$ sink when converted to $pCO_2$ in combination with a second marine $CO_2$ system variable such as DIC or TA. While float-based observations for both DIC and TA are still lacking, and as TA values are readily predictable thanks to established algorithms (e.g., the Locally Interpolated Alkalinity Regression (LIAR) method; Carter et al., 2018) and also less impacted by biological variations (Zeebe and Wolf-Gladrow, 2001), TA is the parameter of choice to derive $pCO_2$ values. Current understanding (e.g., Carter et al., 2018) is that TA can be predicted with a typical uncertainty of about 6 $\mu$mol kg$^{-1}$ which does not include, however, potential regional biases due to insufficient data coverage, contributions from inorganic nutrients as well as biases due to unknown organic TA contributions in highly productive and/or coastal waters. Using this TA uncertainty, u(TA), we calculated the minimum required pH uncertainty, u(pH), that allows to meet two $pCO_2$ uncertainties, u($pCO_2$), as defined by Newton et al. (2015): the "climate goal" uncertainty of 2 $\mu$atm and the "weather goal" uncertainty of 10 $\mu$atm.

To frame the "weather goal", pH uncertainties of around 0.01 pH units (from 0.008 to 0.016 depending on T and $pCO_2$), have to be reached while to derive $pCO_2$ data with an uncertainty as the one defined by the "climate goal" criterion, and considering a u(TA) equals to $\pm$ 6 $\mu$mol kg$^{-1}$, a pH uncertainty < 0.006 pH units is required. At u(TA) = 6 $\mu$mol kg$^{-1}$, the overall contribution of this parameter to the derived uncertainty in $pCO_2$ is rather marginal in comparison to the dominant impact of u(pH), and the resulting $pCO_2$ change represents slightly more than 16% of the pH impact when considering a 0.006 pH units pH-uncertainty. Expressed differently, it means that the uncertainty in predicted TA corresponds to an uncertainty in pH of about 0.001 pH units. However, while the u(TA) is not the major obstacle to derive accurate $pCO_2$ data, TA values still would have to be carefully estimated to then be used as a predictor variable. Regional and/or seasonal biases in estimated TA can be observed in some oceanic regions where high surface nutrient concentrations can occur, especially during phytoplankton bloom situations. The TA-uncertainty can also be more important in areas subject to terrestrial discharges as allochthonous matter or organic TA can be associated with non-carbonate organic alkalinity (Soetaert et al., 2007; Hunt et al., 2011). This perhaps warrants specific tests on the accuracy of TA predictions in critical regions (or seasons) but also if this parameter aims to be used to derive other parameters of the $CO_2$ system, especially DIC. Finally, an additional source of uncertainty when

calculating $p$CO$_2$ (pH, TA) from floats originates from uncertainties in the carbonate system equilibrium constants (Orr et al., 2018).

In order for float-pH data to be suitable for the calculation of parameters of the marine CO$_2$ system, and in particular $p$CO$_2$ data, with useful accuracies, the documented shortcomings in accuracy of float-pH need to be explored and addressed. Taking into account the error propagation, the u(pH) allowed for calculating $p$CO$_2$ from the pH and TA is on the order of 0.0107 $\pm$ 0.0018 for the "weather goal" and 0.0056 $\pm$ 1.42×10$^{-4}$ for the "climate goal". In the SNWA region, the demonstration done in Section 3.1 of this study has shown that the combination of uncertainties associated with the choice of the reference method and reference depth as well as the choice of method to calculate the adjustments for the individual float cycles can lead to uncertainties in pH well beyond what is deemed acceptable to exploit the pH data for CO$_2$ calculation purposes. Thus, to achieve the required $p$CO$_2$ uncertainty, it is desirable to reduce and better constrain the uncertainty associated with float-based pH measurements to derive and depict entirely the oceanic carbon cycle.

## 4   Conclusions

For correcting float-based pH measurements, the current standardized routines from Ago data management rely on a single-point at-depth correction method along with reference algorithms such as LIR-pH, ESPERs or CANYON-B, assuming that the adjustment calculated at-depth yields corrections applicable to the entire profile.

By using both float-based pH data and in situ pH data from other platforms acquired in the SNWA area, this study was able to identify uncertainties and potential biases associated with the adjustment applied which raise concerns about the single at-depth correction on adjusted pH data. Our findings show consistent results indicating that corrected float-pH data may be biased by several hundredths of a pH unit near the surface in the SNWA possibly in response to deep convection events, suggesting that similar observations might be possible in other deep convection regions. Even if the statistical significance of our findings is limited due to the low number of comparisons available, this apparent weakness of the DM QC process of float-pH data should be considered in light of the challenges in interpreting TA and pH-derived $p$CO$_2$ data in a crucial area for ocean convections events and anthropogenic carbon storage. With regards to the situation observed in the SNWA, we suggest (1) to revisit the temperature and pressure effect on the sensor, and (2) to consider global crossover analysis between float-pH surface data and other platforms (SOOP lines, buoy, floats) to independently quality controlled and perhaps correct float-pH data close the surface, where the accuracy required to better constrain the oceanic response to climate changes is the highest.

*Data availability.* Data from the German-SOOP *Atlantic Sail* line are available at https://meta.icos-cp.eu/resources/stations/OS_NA-VOS. Argo data are available at http://doi.org/10.17882/42182#96550 or at ftp://ftp.ifremer.fr/ifremer/argo/dac/coriolis. These data were collected and made freely available by the International Argo Program and the national programs that contribute to it (https://argo.ucsd.edu,https: //www.ocean-ops.org). The Argo Program is part of the Global Ocean Observing System. Data from the MSM94 cruise (https://doi.org/10. 48433/cr_msm94) can be found on the Pangea website (https://doi.pangaea.de/10.1594/PANGAEA.927311).

*Author contributions.* CW-R, TS, and AK initiated and designed the study. TS, and AK helped supervising the study. BK and HB helped revising the manuscript and providing significant inputs. CW-R, TS, and A.K. wrote the first draft of the manuscript. All the authors contributed to manuscript revision, read and approved the submitted version.

*Competing interests.* The authors declare that they have no known competing financial interests or personal relationships that could have appeared to influence the work reported in this paper.

*Acknowledgements.* This work has received funding from the European Union's Horizon 2020 Research and Innovation Program in the Euro-Argo RISE project (grant agreement n°824131). It was further supported by the projects DArgo2025 (FKZ 03F0857A+C) and C-SCOPE (FKZ 03F0877A+B) of the German Ministry for Research and Education. We would like to thank Johannes Karstensen, chief scientist of cruise MSM94, and his team for wonderful cooperation in the context of the achieving a crossover with one of our floats. The two referees are thanked for helping improve this paper.

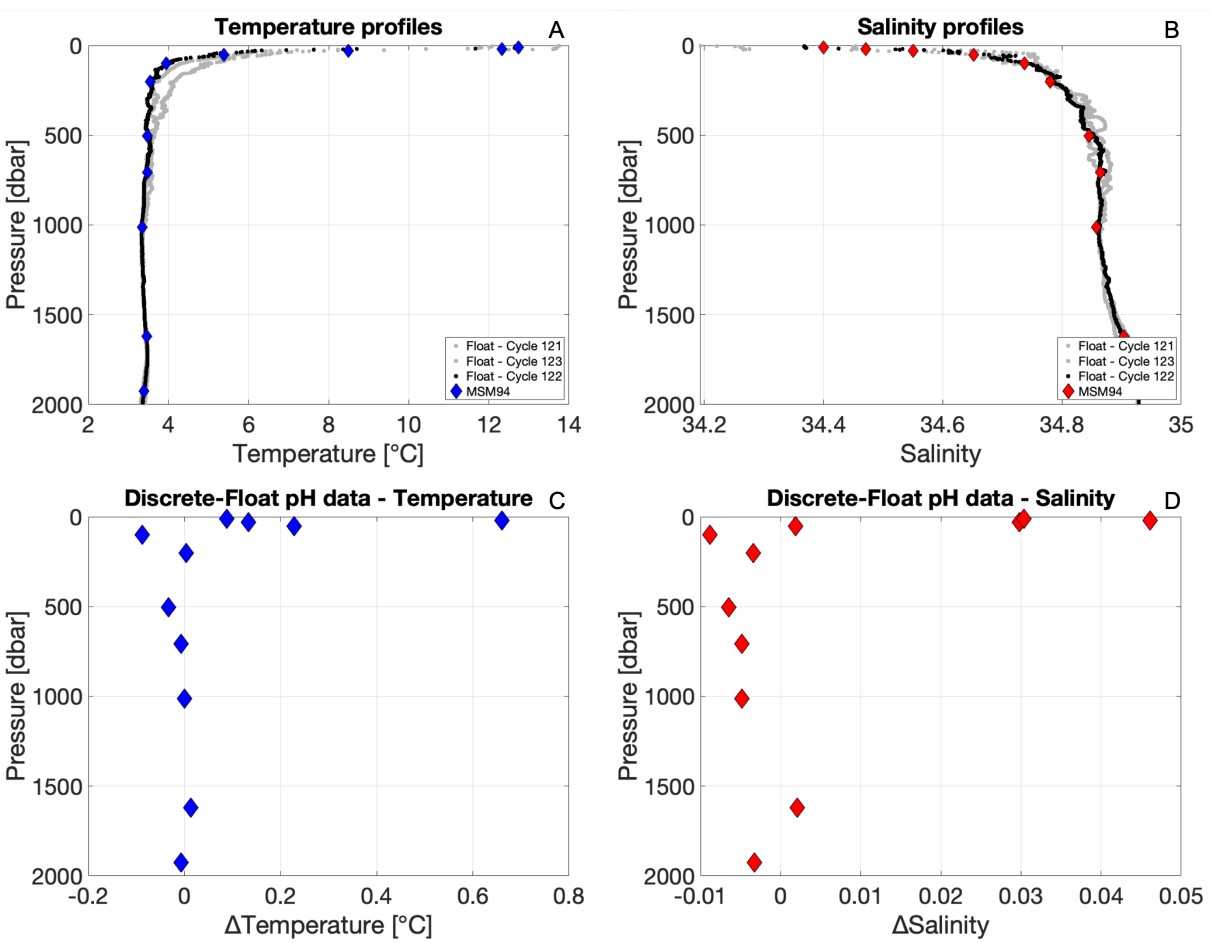

**Figure A1.** (A and B) Vertical profiles of temperature (A) and salinity (B) measured during the MSM94 cruise (diamond dots) and acquired by the float WMO 3901669 during cycles 121, 122 and 123 (gray and black lines, respectively). (C and D) Differences between discrete and float (cycle 122) temperature (C) and salinity (D) data calculated after matching in density space to avoid biases from internal waves.

**Table A1.** Mean differences between raw float-pH data and float-pH data corrected at two distinct depths and using the four different methods for the floats WMO 3901668 and 3901669 only for cycles acquired when the mixed layer depth was deeper than 1000 db. SD stands for Standard Deviation.

| | | Float WMO 3901668 | | Float WMO 3901669 |
| --- | --- | --- | --- | --- |
| | | *Winter 2019* | *Winter 2020* | *Winter 2019* |
| | | *Mean MLD=1639.6 db* | *Mean MLD=1712.1 db* | *Mean MLD=1240.2 db* |
| **ESPER-NN** | *Raw-1500 db* | $-0.0293 \pm 1.39\times10^{-4}$ | $-0.0248 \pm 8.12\times10^{-5}$ | $-0.0352 \pm 9.97\times10^{-5}$ |
| | *Raw-1950 db* | $-0.0181 \pm 5.23\times10^{-4}$ | $-0.0164 \pm 1.08\times10^{-4}$ | $-0.0265 \pm 5.95\times10^{-5}$ |
| **ESPER-LIR** | *Raw-1500 db* | $-0.0416 \pm 3.52\times10^{-5}$ | $-0.0380 \pm 1.64\times10^{-4}$ | $-0.0456 \pm 1.05\times10^{-4}$ |
| | *Raw-1950 db* | $-0.0244 \pm 3.30\times10^{-3}$ | $-0.0300 \pm 1.80\times10^{-3}$ | $-0.0307 \pm 1.77\times10^{-4}$ |
| **CANYON-B** | *Raw-1500 db* | $-0.0508 \pm 8.89\times10^{-5}$ | $-0.0478 \pm 1.41\times10^{-4}$ | $-0.0554 \pm 6.49\times10^{-5}$ |
| | *Raw-1950 db* | $-0.0391 \pm 1.50\times10^{-3}$ | $-0.0399 \pm 8.88\times10^{-4}$ | $-0.0457 \pm 6.20\times10^{-5}$ |
| **LIR-pH** | *Raw-1500 db* | $-0.0677 \pm 4.31\times10^{-5}$ | $-0.0647 \pm 3.95\times10^{-5}$ | $-0.0728 \pm 8.51\times10^{-5}$ |
| | *Raw-1950 db* | $-0.0549 \pm 4.03\times10^{-4}$ | $-0.0543 \pm 2.77\times10^{-4}$ | $-0.0628 \pm 6.18\times10^{-5}$ |

**Table A2.** Statistics of the crossover analysis for SOOP- and float-pH data. N stands for the number of values used to derive the statistics. Crossovers were calculated for $\Delta x \leq 400$ km, $\Delta t \leq 7$ d, and $\Delta S \leq 0.5$. pH values were recalculated using CO2SYS (van Heuven et al., 2011) to account for any temperature difference between matched observations. Float-pH data have been corrected with the SAGE tool using either the reference depth level 950 dbar or 1950 dbar and ESPER-LIR as reference.

| | | | $\Delta$pH at $\Delta T=0$ | | $\Delta$pH at $\Delta S=0$ | |
| --- | --- | --- | --- | --- | --- | --- |
| Correction Depth | N | *Float WMO* | pH offset | Uncertainty of the offset | pH offset | Uncertainty of the offset |
| 950 | 11 | *1902303* | 0.025 | 0.010 | 0.018 | 0.010 |
| 950 | 11 | *1902304* | 0.018 | 0.006 | 0.012 | 0.006 |
| 1950 | 5 | *4903365* | 0.030 | 0.002 | 0.036 | 0.004 |
| 1950 | 2 | *6904241* | 0.042 | 0.0003 | 0.030 | 0.0003 |
| 1950 | 11 | *6904112* | -0.031 | 0.014 | -0.029 | 0.008 |
| 1950 | 8 | *3901669* | -0.004 | 0.007 | 0.013 | 0.006 |

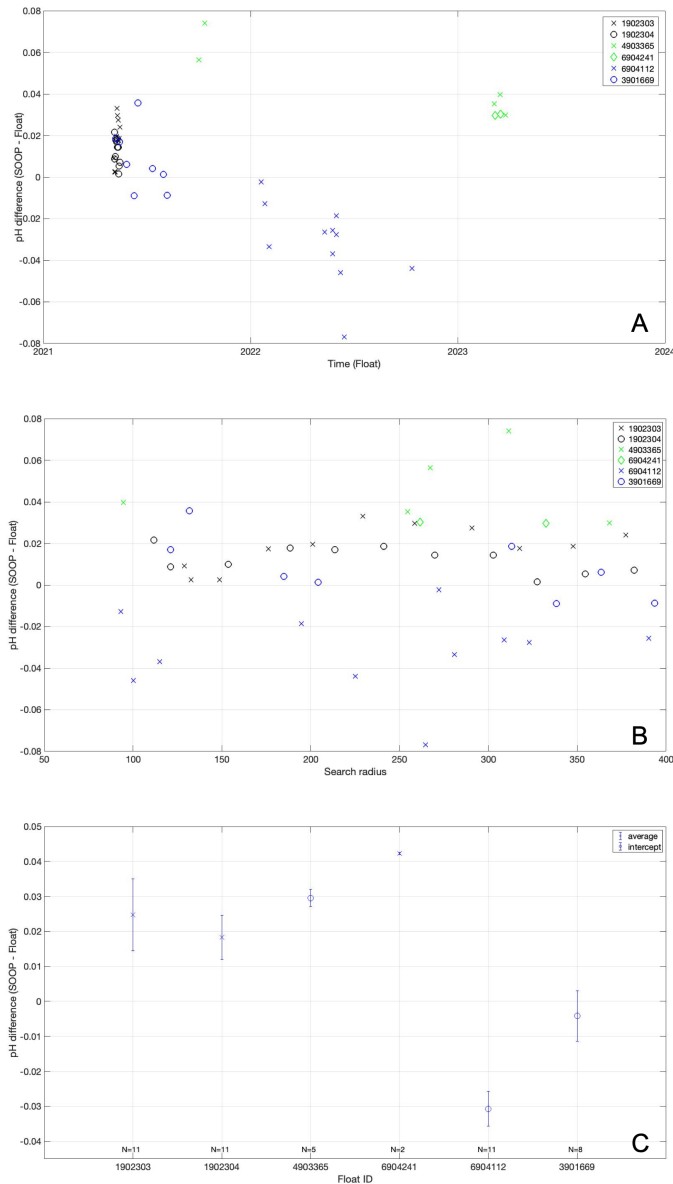

**Figure A2.** Offsets between SOOP-pH and fully corrected float-pH data (y-axis) as a function of the time (panel A) and the crossover criterion (panel B) for the 6 floats considered. Panel C shows the mean offsets and their associated uncertainties. The pH offset was determined at $\Delta T$ = 0 °C (temperature difference between float data and SOOP data) by fitting a linear regression to the data for the float showing a clear spread $\Delta T$ values (dots) or by considering the mean pH difference when clustering around $\Delta T$ = 0 (crosses). Crossovers were calculated for $\Delta x \leq 400$ km, $\Delta t \leq 7$ d, and $\Delta S \leq 0.5$. pH values were recalculated using CO2SYS (van Heuven et al., 2011) to account for any temperature difference between matched observations. Float-pH data have been corrected with the SAGE tool using either the reference depth level 950 dbar or 1950 dbar and ESPER-LIR as reference (see Table A2). N stands for the number of values used to derive the statistics.

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
