# Peer review of "Technical note: Assessment of float-pH data quality control methods - A study case in the subpolar northwest Atlantic Ocean"

_Biogeosciences, 2023_

## Author Comment (AC1)

**Technical note: Enhancement of float-pH data quality control methods: A study case in the subpolar northwest Atlantic Ocean**

*In this response, the "original manuscript" refers to the first submitted manuscript that has been evaluated by the reviewer and the "revised manuscript" refers to the manuscript that has been modified according to the reviewer´s comments. Comments from the reviewer are pasted below in black font; our point-by-point responses immediately follow in blue font. Blue italic sentences are those that have been modified / added in the revised manuscript. When indicated, line numbers refer to the new version of the manuscript (Latex version). We have also submitted a Word document with the "track changes" function activated, which should help the reviewer in figuring out the changes.*

**Responses to REVIEWER 1 - Brendan CARTER**

Wimart-Rousseau et al. have put together an interesting new data set and float-cruise comparison. They also review float adjustment algorithms, adjustment reference depths, and choices regarding how and when adjustments are updated. Their findings show significant offsets between their floats and discrete samples collected from a nearby cruise. They arrive at a plausible set of conclusions and suggest several useful measures. The resulting paper could be a useful contribution to the literature, but I believe it should nevertheless be returned to the authors for revision for the following reasons:

We thank the reviewer for his constructive and helpful comments about our paper, his positive opinion concerning the interest of this manuscript for the community and the time he spent reading and reviewing our manuscript. We most appreciate his relevant comment about the recent global pH algorithm update which has been added in the revised manuscript. Comments and suggestions made by the referee on text and/or figure/table edits are discussed below and addressed in the revised version of our manuscript.

1. This is too long to be a technical note, which, to my read on the journal policies, is limited to "a few pages." This is 21 pages (before references) in the review format. Also, too much of the discussion is qualitative rather than quantitative and did not "feel" technical. This paper needs to be re-worked as a full length paper or shortened and focused.

Following the referee's recommendations and comments listed hereafter in this review, we modified several paragraphs of the manuscript by removing sentences (e.g. Section 3.3) and clarifying some others. Figure 6 has been reduced and Figure 8 removed to simplify the reading. See also our point-by-point responses indicating the changes done. Overall, we accepted all the suggestions proposed by the referee. We believe that these modifications improve the readability of the manuscript.

This paper aims to describe and discuss the main limitations and uncertainties associated with the current float-pH data correction procedure as well as to propose a way forward to enhance the float-pH quality control process. According to the Biogeosciences guideline, we believe that our manuscript is relevant for publication as a technical note as it presents "novel aspects of experimental and theoretical

methods and techniques which are relevant for scientific investigations within the journal scope". For this type of manuscript, no clear indications about the length are given on the webpage, except a "few pages". In its revised form, the manuscript is now 20 pages. As a comparison, a technical note of 18 pages (before references) has been published in 2021 (Canning et al., 2021[1]). We thus believe that the revised manuscript could be published as a technical note too.

2. The discussion about adjustment update methodologies was ultimately unconvincing. I feel it could be reduced to a quick comment that it would be useful to have a community consensus regarding how this is done, which I don't feel needs much justification. Alternatively, the authors could rework and/or expand upon their rationale for their preferred method in a full paper.

In the original manuscript, a discussion about the impact of the correction method used to correct float-pH data was presented in Section 3.1.3. and Figure 5 aimed to illustrate our presentation. The purpose of this section was to discuss the noticeable step-like changes observed with the current correction procedure (i.e., the SAGE method) and to find the best way to represent the smooth sensor drift over time, as observed when looking at the pH time-series recorded at the parking depth (previous Figure A1 in the Supplementary Material). Indeed, in comparison with the pattern of the cycle-by-cycle correction, the high pH changes of ca. 0.01 pH units observed between linear drift phases with the SAGE method appear to be unrealistic. In our view, the sensor rather shows undulations in response with smooth and less smooth phases. This statement is somehow confirmed by the pH sensor behavior when the float drifts at its parking depth. In consequence, we believe that an adaptation of the current correction procedure could be done to better maximize the smoothness of the corrections and to avoid introducing artificial jumps. This presentation could be thus useful for the community and discussion concerning the current procedure could arise from it.

However, we agree that explanations were missing in the original manuscript and that the original idea to put Figure A1 in the Appendix was not relevant as it is critical for our argument. Following the recommendation of the reviewer, Section 3.1.3 has been modified, explanations have been added, and Figure 5 re-drawn: now 4 panels representing differences between raw and corrected float-pH data following the SAGE method (panel A), the GEOMAR methods (panel B), pH data measured at the parking depth (panel C) and pH data measured at the parking depth minus reference (CANYON-B pH data, panel D) are presented. Figure A1 has been modified and is now included in the revised manuscript.

3. The quantitative aspects seemed, in places, potentially incorrect. Other float-to-pCO2 comparisons have not shown as large of offsets at the surface as are found in this study (see discussion in https://agupubs.onlinelibrary.wiley.com/doi/full/10.1029/2022AV000722, also confirmed by some unpublished community studies, and compare to the pCO2 offset implied by the pH offset observed herein of ~50 uatm). Worryingly, the correlation shown herein of the upper-ocean discrete-to-float pH offset to the temperature is comparable in magnitude to the sensitivity of pH itself to temperature, so I would urge a double check on the pH temperature adjustments made in this comparison. If they

[1] Canning, A., Fietzek, P., Rehder, G. and Körtzinger, A. (2021). Technical note: Seamless gas measurements across the land–ocean aquatic continuum – corrections and evaluation of sensor data for $CO_2$, $CH_4$ and $O_2$ from field deployments in contrasting environments. Biogeosciences, 18, 1351–1373. https://doi.org/10.5194/bg-18-1351-2021

were performed correctly, then the calibration of the pH probe sensitivity to temperature seems to be nearly 100% in error (which might be the author's point, though it is odd then that this is not seen elsewhere). The author's point stands that the North Atlantic is a worst case scenario in many respects is accurate, but that it is not an especially problematic location from the perspective of vertical temperature gradients. The authors' comment regarding the differences between the algorithm estimates is supported by a listed standard deviation of the algorithm differences of 0.051… for reasons given in line-by-line comments I believe this represents an order of magnitude error in the listed value or an indication that the standard deviation is not the appropriate statistic in this case. If my suspicions here are correct, and the true standard deviation is <0.01, as it appears from the histogram, then this is roughly consistent with the published algorithm uncertainties at depth when comparing two (only arguably independent) algorithm estimates. This then would give us no reason to doubt the earlier Williams et al. uncertainty propagation.

In the open ocean, we agree with the referee and the current literature that float-derived $pCO_2$ (pH, TA) estimates have a theoretical uncertainty of ~11 µatm. We also thank the referee for suggesting the paper written by Bushinsky and Cerovčki (2023). Notwithstanding, in the studied area, two crossovers comparisons have been performed using two independent datasets and on two different floats. Despite the limited number of floats and crossovers associated with this study providing only one showcase, and although the actual pH values may be slightly different due to the regional variability, the preliminary results point at unacceptably high and almost identical biases in surface pH values from the 2 floats which have been corrected in the exact same way. It calls for an additional independent pH reference in the surface ocean. Indeed, in both cases, pH offsets are positively correlated with temperature, being the smallest at the temperature of the reference depth. In agreement with the referee, and as it is stated in the manuscript, we argue that it points towards an imperfect representation of the temperature and/or pressure dependencies of the pH sensor (Page 16). Our findings show that corrected float-pH data may be biased by several hundredths of a pH unit near the surface in this deep convection region, suggesting that an adjustment of the reference depth might be done in such oceanic areas. Finally, it could also be pointed out that a current publication from Gattuso et al. (2023)[2] presents even higher pH uncertainties for two SeaBird pH sensors deployed in the high-Arctic fjord (Kongsfjorden, Svalbard).

Concerning the pH temperature adjustments: Discrete pH measurements used for the comparison have been converted to in situ temperature and pressure for this study. It has been done using the CO2SYS software with measured pH data and TA values as input variables. Converted discrete pH data have been re-checked for this review and no conversion mistake has been found.

Concerning the standard deviation of the algorithm differences: As stated in our answer to one specific referee's comment, the standard deviation value given in the manuscript is the right one. This value is large compared to the 5th/95th percentiles because of very wide tails in the distribution. Indeed, this wide tail distribution and important standard deviation value are located, for some parts, in the Black Sea causing quite a consistent (and high) difference of ca. -1.6 pH units. In our case, we agree with the referee that this noticeable standard deviation value implies/indicates that the distribution is not Gaussian and raises the question of the utility of this metric considering that percentiles give a more

[2] Gattuso, J.-P., Alliouane, S., and Fischer, P. (2023). High-frequency, year-round time series of the carbonate chemistry in a high-Arctic fjord (Svalbard). Earth System Science Data, Earth Syst. Science Data, 15, 2809–2825. https://doi.org/10.5194/essd-15-2809-2023

robust accuracy assessment metric than standard deviation. In this study, we still wanted to present this value as it is a metric commonly used but also because it reflects one of the main messages of the manuscript: differences are much larger than we would expect from a comparison.

4. A somewhat recent global pH algorithm update (ESPER-LIR and ESPER-NN) were omitted from the analysis. Presumably this is because they are not yet included in SAGE but including them should nevertheless be helpful for this discussion because the updated algorithms take a different approach to several of the algorithm limitations discussed in this paper and are slated to become incorporated into SAGE (to my understanding). For the algorithms that were used, important information was not provided (that I saw) regarding the use of an optional adjustment for ocean acidification. Also, it is arguably appropriate to keep the "spectrophotometric" pH adjustment (as the authors chose to do), but it should be pointed out that this limits the comparability of the two algorithms that were considered (unless such an adjustment was applied to CANYON-B estimates?). As a note here, we (Ocean Carbonate System Intercomparison Forum) are drafting a paper that is taking a step back from the recommendation that this adjustment should always be applied given the finding that the apparent slope is not present in a subset of cruise measurements made with purified dye. The recommendation is to instead treat uncertainty in the comparability of pH and DIC + TA as a component of uncertainty in whatever calculation is being attempted... this recommendation would support the overall thesis of this paper that we need to be cautious about the uncertainty that we claim we can achieve from float-based pCO2. A related comment that this paper uses the term "correction" when in many cases I believe "adjustment" should be used. The distinction I draw is when we believe we understand the reason for the apparent offset it is appropriate to call it a correction, and I would use adjustment in all other cases.

We agree with the referee that a comparison with the new ESPERs methods was missing in the original manuscript. Following the comments and suggestions of the reviewer, the revised manuscript now includes a presentation of pH data estimates with the ESPERs routines. Figures and tables have been subsequently adjusted. In particular, Sections 3.1.1. and 3.1.2. have been modified substantially. Precisions about the optional ocean acidification adjustment have been added too, both in the main manuscript and in figure legends.

Concerning the spectrophotometric pH adjustment: As stated by the referee and precise in the manuscript, this adjustment has been kept in this study. Thus, CANYON-B pH data have been adjusted to align estimates with spectrophotometric pH measurements made using purified dye. As the LIR-pH training dataset consists of values either measured or calculated but adjusted using the same purified-dye adjustment (adjustment 3; Carter et al., 2018), we consider that the two algorithms' results are comparable. In the revised manuscript, a sentence has been added to more clearly state that this adjustment has been used.

Finally, we have paid more attention to the nomenclature and the meaning of the word adjustment in the revised manuscript.

5. Some of the figures are missing information, and the writing is difficult to understand in a few places (but excellent in many other parts). Some of the notation is inconsistent (see line by line comments). There is a comparison to "weather" and "climate" quality

data thresholds, which I argue below is inappropriate. The discrete pH measurement uncertainty is unrealistically low.

In the original manuscript, labels for Figures 4C and 6D were incomplete and explanations about some numbers presented were missing (e.g. Figure 5). Following comments by the reviewer, several figures have been modified in the revised manuscript and explanations added in the legends. All the writing comments and modifications proposed by the referee have been adressed.

Concerning the "weather" and "climate" goals comparison: We agree with the referee that climate and weather goals are related to precision and that the presentation done in the original manuscript could be confused as this discussion occurred after the comparison between corrected float-pH data and in situ discrete measurements (Section 3.2.). On the other hand, Section 3.1. presents a detailed description of the dispersion of the corrected float-pH data in response to the reference pressure choice, the reference depth selection as well as the choice of the method used to correct float cycles. In our manuscript, we believe that, whereas Section 3.2. aims to assess the accuracy of the correction procedure and to discuss the errors, uncertainties presented in Section 3.1. are relevant and allow comparison against the GOA-ON goals. In the revised manuscript, Section 3.3 has been shortened and clarified.

Concerning the pH uncertainty: During the MSMS94 cruise, samples were poisoned onboard following the current standard procedure (SOP) and analyzed at GEOMAR. pH measurements were tested regularly against CRM reference samples to check the accuracy of our measurements. While CRMs are certified only for AT and DIC, pH measurements were also performed for each bag by Dickson's lab and made available for us. The resulting uncertainty in pH measurements for discrete water samples was $\pm 0.002$ pH units. In the studied area, as all the best practice recommendations have been followed, we have no reasons to doubt the resulting pH uncertainty.

Ultimately I was left uncertain of what to make of the results. This is perhaps a useful outcome for a paper that is arguing that we need to be less optimistic about the quality of the data generated by certain approaches, but I feel like the authors should take advantage of a revision to address the issues noted above and below, should double check for potential errors in the presentation and the analysis (particularly the pH- temperature and pressure conversions), should more rigorously compare how the algorithm estimate variability compares to the expected variability in a statistical sense, and should shorten the paper and distill the main ideas (particularly if it is to be kept as a technical note). That said, I tend to agree with all of the conclusions made by the authors, and their presentation did seem to support the conclusions. The subject matter is important and the statements in the conclusions are worth making to the community, so I hope the authors resubmit this paper.

We thank the referee for his insightful and stimulating recommendations regarding our paper. In addressing his concerns, we think that the revised manuscript has been improved substantially. Overall, we have accepted all the suggestions proposed by the referee, significantly modifying the main text and tables, and adding new information in the discussion section.

*Line-by-line comments*

1: "this" refers to a subject that has not yet been defined

The sentence has been modified as follows: "*Since a pH sensor has become available that is suitable for demanding autonomous measurement platforms, [...]*" (L.1).

6: "decipher punctual events" is awkward phrasing. Suggest "Measure the impacts of short-term events"

The sentence has been modified as suggested by the referee. (L.6).

8: This is a matter of personal preference so please feel free to ignore this comment, but I feel this sentence is written backwards. It is shorter and easier to read when written as, e.g., "Quality control is needed to correct sensor offsets or drifts." A recommendation a past advisor gave me is to establish the subject and verb of a sentence early in the sentence so the readers know what the sentence is about. The subject and verb are among the last words in this sentence, and this is a common element of many of the most difficult sentences in this paper.

We thank the referee for this comment and his suggestion. The sentence has been clarified accordingly: "*In consequence, a consistent and rigorous quality-control procedure has been established to correct sensor offsets or drifts as the interpretation of changes depends on accurate data.*" (L.7).

9: Again, this feels backwards

This sentence has been modified as suggested by the referee in his comment #8 (L.7).

12: LIRPH should be LIPHR or LIR-pH

The acronym LIRPH has been replaced in the manuscript as well as on all the figures by LIR-pH.

32: This sentence is correct, but it is probably better to say "ocean acidity will increase" because "alkalinity" will not decrease by this mechanism.

This sentence has been modified as suggested: "*Depending on emission scenarios, ocean acidity will increase with a projected pH decline ranging from 0.16 to 0.44 pH units by 2100 [...].*" (L.32)

49: The majority of what we know about surface ocean pCO2 arguably comes from ships of opportunity, which deserve greater mention in this discussion.

The description of the SOOP network previously done in Section 2.2. of the original manuscript has been shortened and explanations about the SOOP program and its interest added in the revised manuscript.

L.49: "*Since the 1990s, the Ship Of Opportunity Program (SOOP; Goni et al., 2010) aims to obtain data from autonomous instrumentation installed on volunteer merchant ships regularly crossing certain areas. This network contributes to building sustained carbon observing datasets and complements the limited capacity of classical observational strategies as the standard-SOOP framework features, at least, routine pCO₂ observations (e.g. Lüger et al., 2004). In the Atlantic Ocean, parts of the SOOP network are operated in the European Research Infrastructure 'Integrated Carbon Observation System' (ICOS) and the 'Surface Ocean CO₂ Reference Observing Network' (SOCONET).*"

58: Maurer et al. show large pH sensor adjustments, suggesting the pre-deployment calibration is not a major factor since it is seldom used… except perhaps for characterizing the dependence on, e.g., pressure.

We agree with the referee that the pre-deployment calibration is not a major factor with regard to the recent literature results. Nevertheless, these sentences aimed to present the regular and general procedure to follow to obtain reliable and consistent data. Indeed, each sensor system should be processed following a calibration scheme ensuring and demonstrating unequivocally accurate pH determinations.

71: Missing ESPER-NN and ESPER-LIR (which are updates to LIR).

This precision has been added in the revised manuscript: "*Recently, two Empirical Seawater Property Estimation Routines (ESPER; Carter et al., 2021) have been included in SAGE as reference methods. The ESPER-NN method generates estimates from neural networks while the ESPER-LIR routine is based on locally interpolated regressions.*" (L. 77)

98: the "claimed" pH accuracy…

The sentence has been modified as suggested. (L. 107)

104: worth pointing out that this is a single point calibration that doesn't reflect the conditions often found at the reference adjustment depth for pH.

We agree with the reviewer that the uncertainty given was the most optimistic one. We toned down this assessment in the revised manuscript.

L. 113: "*A stringent referencing and adjustment process for the oxygen can yield accuracies around 1.5 µmol kg$^{-1}$ (Bittig et al., 2018a), although depending on the details of the optode calibration, handling, and usage scenario, the accuracy of O$_2$ measurements can vary considerably.*"

120: the GLODAP data are not directly used in the float pH correction procedure in most instances.

This sentence has been shortened: "*In situ pH data measured from water samples are generally considered as reference data for float-based observations and are useful tools to independently estimate pH data accuracy and, if needed, apply additional adjustments.*" (L. 132).

135: this is not a CRM for pH, and there is no certified value

We agree that this is not certified reference material (CRM) for pH. However, our logic is that the CRMs from the Dickson lab are known to be stable for carbon parameters (and thus for pH). We believe that the pH value from the Dickson lab is accurately determined even though it is not a real certification. The uncertainty in the reported pH value for the Dickson CRM will be an order of magnitude less than the difference we observed between the pH data from the SOOP line and the floats. Nevertheless, by moving the sentence line 152 in the original manuscript right after this statement in the revised manuscript (L. 152), we believe that it clarifies the situation. The certified value (7.8417 ± 0.0014 at 25°C) has been added too (L. 153).

137: on what basis is it assigned this very low pH uncertainty given that the best methods are typically assigned an uncertainty of 0.01-0.007? The uncertainties in the calculations from DIC and TA are also quite large, and the uncertainties from the conversion to in situ conditions are thought to be large and poorly known. This claim might be justifiable if it is expressed in terms of reproducibility, but uncertainty has a more expansive definition than reproducibility.

While CRMs are certified only for AT and DIC, pH measurements were also performed for each bag by Dickson's lab and made available for us. The comparison done against these certified materials leads us to conclude that the resulting reproducibility was ± 0.002 pH units. However, we agree that the term uncertainty might be misleading. The sentence is changed to reproducibility. (L.153)

152: This caveat should go earlier.

This sentence has been moved earlier in the revised manuscript and is now line 152.

200: note

The sentence has been modified.

230: and in

The sentence has been modified.

236: measure pH

The term pH has been added.

250: Figure 4 does not obviously support this statement without further explanation

We agree with the reviewer that the reference to Figure 4 wasn't sufficiently supported in the original manuscript. In the revised version of the manuscript, Figure 4 has been modified and is now better introduced and explained: "*Figure 4 exhibits spatial distributions of estimated pH data at the classical reference 1500m depth level using either LIR-pH (with the OA adjustment), CANYON-B, ESPER-LIR, or ESPER-NN and differences between the estimated datasets with uncertainty between reference algorithms in the order of 0.015 pH units in the SNWA area.*" (L.273).

256: what does the "respectively" refer to? Are these different depths? Estimate methods?

The sentence has been clarified: "*In addition, using the SOCCOM array, Maurer et al. (2021) calculated CANYON-B and LIR-pH pH estimates and observed a larger uncertainty toward the surface compared to 1500 m with mean differences (CANYON-B minus LIR-pH pH data) of -0.025 and 0.001 pH units near the surface and at the 1500 m depth level, respectively.*" (L.281).

Figure 4: You should indicate whether LIR-PH estimates are using the OA adjustment option. I believe SAGE usually omits this adjustment, which might explain why LIR-PH is high relative to CANYON-B in the North Atlantic and low in the North Pacific. That said, as you correctly point out, LIR-pH uses an globally uniform OA adjustment that varies only by density, whereas CANYON-B uses an empirical local fit (that, I argue elsewhere, may erroneously project interannual variability forward and backward in time). Carter et al. 2021

attempt to resolve these issues, and this technical note would be more useful if it also included a comparison with the ESPER routine estimates. These routines are slated to become incorporated into SAGE. (Note, you'll probably still find large differences between ESPER pH estimates and LIR-pH estimates, particularly in the North Atlantic, which appear to be attributable to the omission of depth as a predictor variable from ESPER-pH… i.e. the values from ESPER-LIR are much more similar to LIR-PH (and ESPER-NN) when depth is included as a predictor... future updates to ESPER-LIR will likely have depth as an optional predictor to minimize the discontinuity that we'll see if and when the transition from LIPHR to ESPER-LIR is implemented).

We thank the referee for this remark. We agree with the referee that the OA adjustment is omitted by default in SAGE and we have decided to keep this option off in this study with regard to the limitations associated with the OA adjustment (i.e., LIR-pH assumes fixed OA rates over time). This information has been added in the revised manuscript (L.233, Fig.3 label). Nevertheless, in Figure 4, the OA adjustment option was used in order to clarify the figure and to not overall mean bias because of a few oceanic regions. In the revised manuscript, this precision has been added both in the main text and the caption (L.274, Fig.4 label). The spatial distribution derived without the OA adjustment (Fig. 1 below) presents a higher mean bias of 0.002 pH units (against -0.001 pH units) which is caused, at least partially, by the Black Sea causing quite a consistent (and high) difference of ca. -1.6 pH units. By removing this sea during the simulation, it appears that the std decreases considerably. High differences can also be explained by the both enclosed and undersampled Mediterrean Sea and Baffin Bay.

[Figure]

**Figure 1** (not in the revised manuscript). Spatial distributions of estimated pH data at 1500 m using different reference models: LIR-pH (without the OA adjustment) (A) and CANYON-B (D). The map of the spatial difference between the two estimated pH datasets is presented in panel (B). Panel (C) shows the bias ΔpH distribution (with statistics). The upper colorbar indicates the difference between estimated pH data using the two models and the lower colorbar gives the pH values.

We also agree with the referee that a comparison with the ESPER methods was missing in the original manuscript. Following the comments and suggestions of the reviewer, Figures 3 and 4 have been updated in the revised manuscript and now include a presentation of pH data estimates with the ESPER routines.

L.284: *"The new ESPERs methods attempt to resolve the issues encountered with existing routines (especially the OA estimate) by expanding their functionality and being trained on a larger data product. In comparison with the LIR-pH estimates, large differences are observed in the SNWA region and might be attributable to the OA adjustment as well as the omission of depth as a predictor variable from ESPER-LIR (Carter et al., 2021). Updated global algorithms (i.e., ESPERs) show comparable estimates in the SNWA area with ESPER-LIR pH estimates slightly higher than pH data estimated with CANYON-B or ESPER-NN. In the dynamic and strongly human-impacted studied region, the lack of coordinate information as a predictor variable in the ESPER-LIR routine could also be argued as an explanation of the observed differences. However, according to Carter et al. (2021), regional assessment statistics obtained in the Northern Atlantic indicate almost similar biases for both the ESPERs and the CANYON-B methods, with a better RMSE statistic for CANYON-B*."

Figure 4c: The y axis is not labeled. Also, there may be a missing 0 in the std value?: It is only possible for the STD to exceed the 5th and 95th percentiles if there are a small number of extreme outliers (that should have likely been omitted). This is among the most important numbers in this manuscript, to my thinking. If it is much smaller, then I'd ask whether the std is indeed much larger than we'd expect from a comparison of two algorithm reference adjustments?

We agree with the referee that the y-axis wasn't labeled for Figure 4c in the original manuscript. The original thinking behind this omission was related to the kind of plot presented directly: to our thinking, the relative distribution of a histogram is what is relevant to check, and the absolute number reported as frequency distribution or count is depending on the dataset resolution. In the revised manuscript, Figure 4 has been re-drawn and the y-axis label "Frequency" has been added (Page 13).

The std value given is the right one: The std is as large compared to the $5^{th}/95^{th}$ percentiles because of very wide tails in the distribution. In our case, this noticeable std value implies/indicates that the distribution is not Gaussian and raises the question of the utility of this metric considering that percentiles give a more robust accuracy assessment metric than std. In this study, we wanted to let this std value on Figure 4c as it is a metric commonly used but also because it reflects one of the main messages of the manuscript: differences are much larger than we would expect from a comparison. Nevertheless, as stated in the former answer to the referee, data outside the $5^{th}/95^{th}$ percentile, and explaining this wide tail distribution and important std value, are located, for some parts, in the Black Sea causing quite a consistent (and high) difference of ca. -1.6 pH units. To clarify this figure and not lead to misinterpretation, we have decided to remove this area in Figure 4 of the revised manuscript. The std is then reduced, even if percentiles indicate that there can be quite some deviations, especially due to the Mediterranean Sea and the Baffin Bay.

[Figure]

**Figure 2** (Fig. 4 in the revised manuscript). Spatial distributions of estimated pH data at the classical reference depth 1500 m using different reference models: LIR-pH (with the OA adjustment) (a), CANYON-B (f), ESPER-LIR (k) and ESPER-NN (p). Maps of the spatial difference between the estimated pH datasets are presented in panel (b-d, g-h and i). Panel (e, i-j and m-o) shows the bias ΔpH distribution (with statistics). The upper colorbar indicates the difference between estimated pH data using the different models and the lower colorbar gives the pH values. For clarity, pH data estimated for the Black Sea have been removed for this simulation as they were outside the 5th/95th percentile and they caused a noticeable increase of the standard deviation (std).

261: from the way the

The sentence has been modified. (L.296)

264: is the noise uniform with depth? If not, then it should be handled by averaging the adjustment across a depth range. If it is, then should it be removed by adjusting every profile independently?

The pH sensor can generate occasional and non-uniform spikes due to electrical noise and despiking is appropriate. According to Johnson et al (2022)[3], the default Argo spike test in core variables does not work for pH because of the strong vertical gradient dependency of this test, making it regionally dependent. On the other hand, the spike test recommended for chlorophyll (Schmechtig et al., 2014)[4] is more appropriate for pH. Thus, a data point is considered a spike and marked with quality flag 4 (data bad) if the test value is > 0.04 pH. If float-pH data have a good QC, then they are adjusted and corrected following the current procedure.

266: "corrected for" should be "removed"

The sentence has been modified.

[3] Johnson, K., Maurer, T. and Plant, J. (2023). BGC-Argo quality control manual for pH, in preparation.
[4] Schmechtig, C., Claustre, H., Poteau, A. and D'Ortenzio, F. (2014) Bio-Argo quality control manual for Chlorophyll-A concentration, Version 1, December 17th 2014. IFREMER, 13pp. https://doi.org/10.13155/35385.

Figure 5: The use of 2 y axes is very confusing here and defeats the purpose of being able to compare the adjustments to one another, though the goal of increasing visibility makes sense. It would probably be better to use a single y axis, which would allow the plot to focus in on the 0.04 to 0.06 range. The greater vertical resolution should also improve visibility. If the different methods are difficult to distinguish on this unified scale, then that suggests that the methods aren't different enough to worry about on this scale. What are the numbers in the upper right? (it's not hard to guess, but it is better if it is spelled out)

We thank the referee for this comment and his suggestion. Considering this remark as well as the one about Figure A1, Figure 5 has been updated (Page 14 in the revised manuscript) and presents now 4 panels representing differences between raw and corrected float-pH data following the SAGE method (panel A), the GEOMAR methods (panel B), pH data measured at the parking depth (panel C) and pH data measured at the parking depth minus reference (CANYON-B pH data, panel D). We also agree with the reviewer that there were no explanations about the numbers in the original manuscript. In the revised manuscript, they are now explained in the figure caption.

By splitting previous Figure 5 (A and B) into two separate figures, we believe that the new organization of the figures helps the reader to identify the impact of the sensor drift correction used on the final adjustment. We also agree with the referee that the impact of the correction method on the final corrected dataset is almost non-significant, especially regarding the mean difference values. Nevertheless, as previously stated, this section aims to discuss the better representation of the sensor behavior over time and we believe that, by merging Figure A1 (in the original manuscript) to the original Figure 5 in the revised manuscript, this purpose has been clarified to the reader.

[Figure]

**Figure 3** (Fig. 5 in the revised manuscript)**.** Mean differences between raw float-pH data minus float-pH corrected using the SAGE tool (Fig. 5A), the cycle-by-cycle GEOMAR method (yellow dots, Fig. 5B), and the linear mean regression GEOMAR method (blue dots, Fig. 5B) and the 3-point centered running mean correction method (green dots, Fig. 5B) for float WMO 3901669. In every case, CANYON-B was chosen as a reference method, and 1500 dbar was chosen as the reference depth. Mean differences between raw and corrected float-pH data with the standard deviations are shown in the legend boxes for each reference method. Figure C shows, for comparison with the SAGE correction, the uncorrected pH data measured at the parking depth (right y-axis) with black dots representing mean pH values for each day. The colorbar shows pressure. Figure D shows differences between raw float-pH data minus float-pH corrected using the SAGE tool

(purple dots, left y-axis) and differences between uncorrected mean pH data measured at the parking depth minus mean reference CANYON-B pH data calculated using measurements recorded at the parking depth (red dots, right y-axis).

278: profile-by-profile or cycle-by-cycle? Either is fine but be consistent. Also, doesn't it remove too much sensor noise, or am I misunderstanding?

We thank the referee for pointing out this mistake. The denomination has been uniformized in the revised manuscript and the term "cycle-by-cycle" is now used. Lines 315 and 322 have been modified in Section 3.1.3. Moreover, we agree with the referee that this sentence was confusing in the original manuscript, especially regarding the denomination "sensor noise" used. The line 278 (in the original manuscript) was related to short-term sensor drifts and noises along the water column, which in fact could not be only related to the correction method but also to the variability in the algorithm estimates as well as chemical reactions close to the sensor chip. With regard to these doubts, this unclear and unverifiable sentence has been modified in the revised manuscript. Indeed, the purpose of this section, generally speaking, is only to discuss the noticeable step-like changes observed with the current correction procedure and to find the best way to represent the smooth sensor drift over time.

L.315: "*The cycle-by-cycle adjustment has the disadvantage that it gives discontinuous adjustment rather than a segmented set of piecewise adjustments*".

282: "has to" is an overstatement as this is not currently done by many data centers (this is another case where the verb is among the last words in the sentence). Also, you have not completely made the case against the cycle-by-cycle approach or the SAGE approach. The basis for the SAGE approach, I believe, was based upon the first principles assumption that the reference potential jumps with discrete events and that the wiggles around the jumps reflect variability in the algorithm estimates or short term sensor noise. This hypothesis would need to be discounted to really make the case effectively.

We thank the referee for pointing out this overstatement and suggesting a reformulation of this sentence. In the revised manuscript, this sentence has been modified and tone down as: "*Therefore, the adjustment method should involve techniques such as a higher-order spline fit, a centered running mean, or a segment separation of the record into linear drift phases.*" (L.318)

We agree with the reviewer that there is no clear comparison in the original manuscript. In the revised manuscript, Figure 5 has been modified and explanations regarding our interpretation of the current correction method have been added. Moreover, as stated in one former answer to the referee, the purpose of this section was to discuss the noticeable step-like changes observed with the current correction procedure (i.e., the SAGE method) and to find the best way to represent the smooth sensor drift over time, as observed when looking at the pH time-series recorded at the parking depth. Indeed, in comparison with the pattern of the cycle-by-cycle correction, the high pH changes of ca. 0.01 pH units observed between linear drift phases and leads to step-like changes with the SAGE method appear to be unrealistic. In our view, the sensor rather shows undulations in response with smooth and less smooth phases. This statement is somehow confirmed by the pH sensor behavior when the float drifts at its parking depth. In consequence, we believe that an adaptation of the current correction procedure could be done to better maximize the smoothness of the corrections and to avoid introducing artificial jumps.

*L.327: "The pH sensor behavior when the float drifts at its parking depth is in agreement with this observation (Fig. 5C). In comparison with float-pH data corrected using the SAGE method, no strong visible discontinuities in raw pH data are observed while the float drifts between its measurement's phases. In our view, the sensor rather shows undulations in response with smooth and less smooth phases over time. In order to test the impact of the reference method on the adjustment pattern, differences between uncorrected float-pH data and CANYON-B pH data derived at the parking depth are presented in Figure 5D. Once again, the pH time-series shows smoothed transitions and the general pattern does not present noteworthy jumps. Such sharp transitions can perhaps be best corrected with our modified GEOMAR segment method or alternatively with a spline fit or a 3-point centered running mean (Fig. 5B)."*

293: discontinuities are not a problem for QC. The statistics still work fine. They are perhaps a problem for studies examining biogeochemical variability over time for a specific profiling float, but these studies would also be challenged by excessively smoothed transitions if a reference potential jump occurred. I believe the authors have a strong case to make here, but this presentation leaves me more confused than convinced. I'd urge them to instead focus on the consequences for a common biogeochemical analysis that would be affected by discontinuities (leading to, e.g., discontinuities in DIC vs. time… though even then a clearly visible discontinuity in DIC might be preferable to a smooth-seeming but equally spurious excursion in DIC as the smoothed adjustment factor catches up to the true adjustment factor). These arguments lead me to the belief that the best approach would be a 1000-1500 m average adjustment applied cycle-by-cycle.

We agree with the referee that statistics are fine and almost not impacted by the discontinuities observed depending on which method is used to correct float-pH data. Nevertheless, as now more clearly stated in the revised manuscript, we believe that some corrections methods, especially the cycle-by-cycle and the SAGE linear adjustment ones, induce jumps that are not observed either on float-pH data time-series or when pH data are recorded while the float is at its parking depth. In consequence, we argue that, when a peculiar float-pH profile is used in comparison with discrete pH measurements in order to compare and examine the accuracy of the correction, such artificial variability induced by the method and not related to the sensor itself could lead to biases and possible misadjustment. In the revised manuscript, sentences have been added to clarify our point of view.

*L.338: "Indeed, and even if the impact of the adjustment method on the final corrected dataset is almost non-significant regarding the mean differences values (Fig. 5D), the possible impact of such artificial jumps induced by the method itself rather than the pH sensor could be noticeable if float-pH data related to these peculiar discontinuous cycles are compared against discrete pH measurements and then adjusted (see Section 3.2)."*

289: Figure A1 is critical for your argument. It needs to be brought into the main text if this section is retained in the final manuscript. It needs to be well-explained and your rationale for preferring the smoothed adjustments over alternatives needs to be more strongly and thoroughly defended. The rationale should go quantitatively beyond "In our view, the sensor rather shows undulations in response with smooth and less smooth phases."

We thank the referee for this relevant suggestion which helps supporting the discussion in Section 3.1.3. Figure 5 has been re-drawn in the revised manuscript and includes now the former Figure A1. We also

thank the reviewer for his suggestion to better describe the figure in the main manuscript in order to use it as an argument for our assessment. The legend of the new Figure 5 has been modified accordingly and explanations have been added in the revised manuscript.

L.327: *"The pH sensor behavior when the float drifts at its parking depth is in agreement with this observation (Fig. 5C). In comparison with float-pH data corrected using the SAGE method, no strong visible discontinuities in raw pH data are observed while the float drifts between its measurement's phases. In our view, the sensor rather shows undulations in response with smooth and less smooth phases over time."*

293: others would argue that these are correcting biases of that magnitude

In the studied area, we observed that the current float-pH correction procedure is impacted by the choice of the reference method used to correct the data (uncertainty of ca. 0.015 pH units), the choice of the reference depth (uncertainty of ca. 0.005) but also the method itself used to correct data which could lead to biases of up to 0.01 pH units. In consequence, with regard to the literature stating that for SBE pH sensors, the accuracy ranges from ± 0.05 pH units (manufacturer statement) to ± 0.005 pH units after data adjustment (Johnson et al., 2017)[5], we believe that these discontinuities have to be more constrained to decrease the adjusted error.

Figure A1: would be more useful if pHTotalInsitu were plotted as the difference from the algorithm estimate. Also, it is unclear if the parking depth pH value has any adjustments applied and, if so, on what basis. Finally, the indication of whether pH is "total scale pH" is inconsistent in these figures. Based on other conversations with people who have strong opinions about these things, my recommendation is to universally use pHT to indicate total scale pH.

We acknowledge the reviewer for suggesting a comparison between pH data measured at the parking depth and pH data estimated using an algorithm. This comparison has been added in the revised manuscript using CANYON-B as a reference method. We believe that this new figure (labeled Figure 5D) improves our presentation and highlights well that the pattern observed for corrected float-pH data (ex. Figure 5A) is not related to the reference method used to correct float-pH datasets but rather to sensor drift adjustment done by each method. In Figure 5D (in the revised manuscript), the y-axis label "pHT in situ" has been replaced by "pH raw" as these data are uncorrected. When pH data recorded at the parking depth are plotted, they are uncorrected. This precision has been added in the legend of Figure 5.

L.330: "*In order to test the impact of the reference method on the adjustment pattern, differences between uncorrected float-pH data and CANYON-B pH data derived at the parking depth are presented in Figure 5D. Once again, the pH time-series shows smoothed transitions and the general pattern does not present noteworthy jumps. Such sharp transitions can perhaps be best corrected with our modified GEOMAR segment method or alternatively with a spline fit or a 3-point centered running mean (Fig. 5B).*"
* * *
[5] Johnson, K. S., Plant, J. N., Coletti, L. J., Jannasch, H. W., Sakamoto, C. M., Riser, S. C., et al. (2017). Biogeochemical sensor performance in the SOCCOM profiling float array. J. Geophys. Res. Oceans 122, 6416–6436. https://doi.org/10.1002/2017JC012838.

302: Yikes! Variants on this experiment have been performed by several members of the community, and none, to my recollection, saw consistent differences this large.

We agree with the referee that the literature showing comparisons of quality-controlled float-pH data against shipboard reference data shows differences much lower. For example, Maurer et al. (2021)[6] present a median bottle-minus float difference for pH data of 0.002 ± 0.015 pH units. In the revised study, including the ESPER methods, we found mean differences ranging between -0.0659 and -0.0150 pH units, and varying according to both the reference pressure level and the reference method used to correct float-pH data. Nevertheless, when excluding the LIR-pH method which seems to induce an over-adjustment of the dataset, and when removing the first four measurements (measured within the first 50 meters of the water column), mean differences of 0.009 and -0.0018 pH units are obtained using ESPER-LIR and ESPER-NN, respectively, and the 1950 db reference pressure. By considering the lab-to-in situ pH conversion uncertainty introduced through calibration (0.005 pH units; Williams et al., 2017[7]), lower uncertainties are even obtained. Thus, we believe that the current correction procedure is relevant at depth but that, in this area, large differences are observed near-surface and might reflect an imperfect representation of the temperature dependence. This assumption is at some points confirmed by the comparison between SOOP-based pH measurements and float-pH data pointing towards apparent biases toward the surface. Moreover, in a recent publication focusing on a high-Arctic fjord (Kongsfjorden, Svalbard), Gattuso et al. (2023)[2] present offsets between spectrophotometric reference samples and a calibrated SeaFET pH time series ranging between ± 0.02 pH units.

In the revised manuscript, some details about the uncertainties to consider (i.e., bottle pH inaccuracy and lab to in situ pH conversion uncertainty) have been added to discuss the results observed and tone down the observed differences.

L. 361: "*Moreover, the laboratory-to-in-situ temperature pH conversion uncertainty of 0.005 pH units (Williams et al., 2017), as well as the absolute uncertainty in the bottle pH measurements (here 0.002 pH units), have to be taken into account before drawing strong conclusions.*"

302: There is a growing sense in the community that bottle pH samples are not well preserved even when following SOPs for DIC and AT storage. Is this possibly a discrete sample issue? Do you have some at-sea measurements to compare with?

During the MSMS94 cruise, samples for total alkalinity, dissolved inorganic carbon as well as pH measurements were taken. These samples were poisoned onboard following the current standard procedure and analyzed at GEOMAR. pH measurements were tested regularly against CRM reference samples to check the accuracy of our measurements. While CRMs are certified only for TA and DIC, pH measurements were also performed for each bag by Dickson's lab and made available for us. In consequence, and even if no at-sea measurements are available to compare with, the comparison done against these certified materials leads us to conclude that the resulting uncertainty in pH measurements for discrete samples was ± 0.002 pH units.

[6] Maurer, T. L., Plant, J. N., and Johnson, K. S. (2021) Delayed-Mode Quality Control of Oxygen, Nitrate, and pH Data on SOCCOM Biogeochemical Profiling Floats, Frontiers in Marine Sciences, 8, 683207. https://doi.org/10.3389/fmars.2021.683207, 2021.

7 Williams, N. L., Juranek, L.W., Feely, R. A., Johnson, K. S., Sarmiento, J. L., Talley, L. D., Dickson, A. G., Gray, A. R., Wanninkhof, R. Russel, J. L., Riser, S. C. and Takeshita, Y. (2017). Calculating surface ocean $pCO_2$ from biogeochemical Argo floats equipped with pH: An uncertainty analysis, Global Biiogeochemical Cycles, 31, 591-604, https://doi.org/10.1002/2016GB005541.

302: the two numbers in this figure only make sense after looking at figure 6, and it should be clear from the text alone

In the revised manuscript, Figure 6 is now described in the main text and the numbers, related to the observed differences, better explained.

L.347: "*Figures 6A and B present differences between discrete pH measurements and float-pH data along the water column and according to two distinct reference pressure levels. We find mean differences ranging between -0.0659 and -0.0150 pH units (Fig. 6B) between the reference pH cast and the fully corrected pH of cycle 122, with higher differences found for the "classical" reference depth of 1500 dbar, and the lowest differences reported for the two ESPER methods.*"

6D: the y axis is unlabeled and missing many negative signs (looks like the figure was just cut off on the left). Far more worrying, the delta pH is about the same magnitude as the change in pH expected from changes in temperature. It is unlikely that there is a 100% uncertainty in the calculated pH change with temperature, which leads a possible simpler explanation that the pH vs. temperature correction was performed incorrectly in this comparison. Were the discrete pH values recalculated at the in situ temperature and pressure? When you say "discrete water temperature" is this the temperature at the time of bottle triggering or the laboratory temperature at the moment of analysis?

We thank the referee for pointing out that Figure 6D was incomplete in the original manuscript. In the revised manuscript, Figure 6 (Figure 4 below) has been re-drawn (Page 16 in the revised manuscript) and now presents differences between discrete pH measurements and float-pH data along the water column and according to two distinct reference pressure levels (1500 dbar, Fig. 6A and 1950 dbar, Fig. 6B) and $\Delta$pH (discrete pH measurements minus float-pH data corrected at the reference depth level 1950 dbar, Fig. 6C) as a function of the difference between discrete water temperature and temperature values recorded at the reference depth of 1950 dbar (i.e., 3.3733°C). Here, discrete water temperature refers to the temperature measured in situ at the time of bottle triggering (at sea). This precision has been added in the legend of Figure 6 in the revised manuscript. On every panel of the revised Figure 6, the four reference methods are also presented.

Discrete pH samples were analyzed at GEOMAR right after the cruise at standard temperature (~25°C) and atmospheric pressure and have been converted to in situ temperature and pressure for this study. The conversion was done using the CO2SYS software with measured pH data and TA values as input variables (see Table 1). A double check for potential errors has been done. Thermodynamic calculations within the carbonate system used the carbonic acid dissociation constants of Mehrbach et al. (1973) as refit by Dickson and Millero (1987), the dissociation constant for bisulfate of Perez & Fraga (1987) and Uppström (1974) for the ratio of total boron to salinity.

| pH measured in the laboratory at ~ 25°C | pH measurement - temperature in the laboratory [°C] | TA measured [µmol/kg] | Temp. *in situ* [°C] | Salinity | Pressure *in situ* [dbar] | pH converted to *in situ* Temp. & Pres. |
|---|---|---|---|---|---|---|
| 7,708 | 25,0173 | 2304,48 | 3,3931 | 34,9257 | 1925,8 | **7,9563** |
| 7,706 | 24,9913 | 2303,18 | 3,4606 | 34,9041 | 1620,4 | **7,9647** |
| 7,694 | 25,01433 | 2300,47 | 3,3397 | 34,8582 | 1012,5 | **7,9779** |
| 7,687 | 25,0256 | 2298,7 | 3,4712 | 34,8644 | 706,1 | **7,9805** |

| | | | | | | |
|---|---|---|---|---|---|---|
| 7,681 | 25,0053 | 2296,33 | 3,4765 | 34,8456 | 504,5 | **7,9816** |
| 7,689 | 25,055 | 2296,44 | 3,5466 | 34,7806 | 201,6 | **8,0018** |
| 7,685 | 25,0313 | 2294,06 | 3,94 | 34,7375 | 100,9 | **7,9947** |
| 7,737 | 24,9803 | 2298,01 | 5,3817 | 34,6516 | 51,5 | **8,0283** |
| 7,848 | 24,9823 | 2295,34 | 8,4924 | 34,5503 | 30,6 | **8,0959** |
| 7,875 | 25,03267 | 2293,27 | 12,3241 | 34,4711 | 19,9 | **8,0646** |
| 7,865 | 24,935 | 2272,82 | 12,7456 | 34,3996 | 10 | **8,0466** |

**Table 1** (not in the revised manuscript). Parameter values used as inputs to convert pH data from standard temperature (~25°C) and atmospheric pressure to in situ temperature and pressure using the CO2SYS software. The last column on the right side of the table presents pH data used in this study.

Example: conv=CO2SYS(7.694,2300.47,3,1,34.8582,25.0143,3.3397,0,1012.5,0,0,1,4,2);

conv(:,18)=7.9779 %    18 - pH output

with CO2SYS (pH measured value, TA value, parameter type (pH), parameter type (TA), salinity, temperature input (during the measurement), temperature output (in situ), pressure input (during the measurement), pressure output (in situ), SI concentration, PO4 concentration, selection of the pH scale, selection of the $K_1K_2$ constants, selection of the KSO4 constants).

[Figure]

**Figure 4** (Fig. 6 in the revised manuscript). (A and B) Differences between discrete and float-pH data (for the cycle 122) calculated after matching in density space to avoid biases from internal waves and corrected using corrected reference levels of 1500 dbar (Fig. 6A) and 1950 dbar (Fig. 6B). (C) ΔpH (discrete pH measurements minus float-pH data corrected at the reference depth level 1950 dbar) as a function of the difference between discrete water temperature (i.e., the temperature measured in situ at the time of bottle triggering at sea) and temperature values recorded at the reference depth of 1950 dbar. The color code refers to the reference method used to correct float-pH data: CANYON-B (yellow diamonds), LIR-pH (green diamonds), ESPER-NN (purple diamonds) or ESPER-LIR (bleu diamonds).

324: SOOP-pH is not a mature effort to my understanding. Some comments on the SOOP-pH methods and QC practices are warranted. SOOP-pCO2 comparisons, to my understanding, are showing much more modest implied float offsets

Indeed, most SOOP feature only $pCO_2$ measurements but other $CO_2$ system variables are coming along well. We have put much effort into testing, improving and assessing autonomous spectrophotometric TA measurements (CONTROS HydroFIA TA) and have reached a quite decent accuracy of about 5

µmol kg$^{-1}$ in unattended SOOP mode (Seelmann et al., 2019[8], 2020a[9], 2020b[10]). Using a much simpler analytical setup (as it does not require sample acidification and $CO_2$ stripping) of this commercial spectrophotometric system for pH (CONTROS HydroFIA pH), we have gained quite some experience in SOOP-based pH measurements. Because of the relatively high stability of the pH measurement, a suite of 5-8 repeated CRM-reference measurements are performed in port before and after each 5-week autonomous roundtrip (Fig. 5 below). These pre- and post- calibration runs are rather stable for each meta-cresol purple (mCP) indicator bag. This yields a clear and consistent track of the small pH drift over consecutive roundtrips which allows us to correct the measured pH to CRM values. Given the small standard deviations of the CRM measurements we believe that the SOOP-pH is of about ± 0.003 pH units.

[Figure]

**Figure 5** (not in the revised manuscript). pH measurements performed on CRM batch 190 with the CONTROS HydroFIA pH system before and after each 5-week roundtrip of the DE-SOOP *Atlantic Sail*. Adjustments of measured pH to the nominal pH value assigned to the CRM (7.8417 ± 0.0014 at 25°C) is based on the mean of all CRM measurements carried out per individual meta-cresol purple bag.

328: needs

The verb has been corrected.

Table 2: A standard deviation of 0.5 salinity units is quite large, is it not? Usually this represents an offset of hundreds of km or more in the surface of the North Atlantic except in

[8] Seelmann, K., Aßmann, S. and Körtzinger, A. (2019). Characterization of a novel autonomous analyzer for seawater total alkalinity: Results from laboratory and field tests. Limnol. Oceanogr.: Methods. https://doi.org/10.1002/lom3.10329.

[9] Seelmann, K., Steinhoff, T., Aßmann, S. and Körtzinger, A. (2020a). Enhance ocean carbon observations: Successful implementation of a novel autonomous total alkalinity analyzer on a Ship of Opportunity. Front. Mar. Sci., 7. https://doi.org/10.3389/fmars.2020.571301.

[10] Seelmann, K., Gledhill, M., Aßmann, S. and Körtzinger, A. (2020b). Impact of impurities in bromocresol green indicator dye on spectrophotometric total alkalinity measurements. Ocean Sci., 16, 535-544. https://doi.org/10.5194/os-16-535-2020.

areas of very strong salinity gradients near coasts. A mean of 0.4 is just as worrisome. Am I misreading this?

Crossovers in the surface ocean are much harder to achieve due to the typically large spatiotemporal variability there. This is particularly the case in the subpolar Northeast ocean, where our SOOP and BGC-Argo float measurements take place. The close proximity of the warm and saline waters of the North Atlantic Current and the cold and less haline waters of Arctic origin cause particularly high spatiotemporal variability. The stricter SOCAT criterion would have yielded only very few crossovers with little statistical significance. We therefore decided to enlarge the search window considerably for the sake of yielding more crossovers. Of course, these individual crossovers are per se even less statistically significant. By plotting all delta-pH vs. delta-T between the float and SOOP pH measurements (note that SOOP pH data were corrected to the float pH observations using CO2SYS and the observed SOOP-TA), we were hoping to find a reasonably linear correlation which at delta-T = 0 should yield a relatively robust estimate of the delta-pH. On average the two pH datasets differed by about 1°C for both floats (although in opposite directions). So clearly this is not a perfect match. We did the same for a correlation vs. delta-S which was slightly less well-constrained but yielded essentially the same pH offset. For both floats, we found about the same salinity offset, again in opposite directions. In a possible future approach that harnesses SOOP-pH for assessment/correction of float-pH, further thought should be put into checking and optimizing the crossovers. Still, we are convinced that the results shown despite their limitations and because of their consistency across the 2 floats and with the discrete hydrocast crossover in the Labrador Sea are clear evidence of an accuracy issue with upper ocean float pH. This we feel is important for the community and should foster similar studies.

364: Climate and weather goals are specifically formulated based on needs for characterizing (paraphrased) "changes in carbonate ion concentrations." This means they are related to precision and not accuracy. This analysis is concerned with accuracy to a much greater extent than precision, so this is not an apples to apples comparison. A better comparison would be how the offset varies over time for a given float or varies between two or more floats in the same location.

Recently, the Global Ocean Acidification Observing Network (GOA-ON) has discussed measurement quality goals that need to be met to ensure appropriate quality to address the relevant problems. Thus, GOA-ON has proposed two key goals corresponding to two levels of related uncertainties: the weather and the climate goal. We agree with the referee that climate and weather goals are related to precision, i.e., "the result of a measurement that permits a statement of the dispersion (interval) of reasonable values of the quantity measured, together with a statement of the confidence that the (true) value lies within the stated interval" (Newton et al., 2015[11]). In our manuscript, whereas Section 3.2. aims to compare float-pH data against discrete pH measurements to assess the accuracy of the correction procedure (and then discuss the errors of the corrected datasets), Section 3.1. presents the dispersion of the corrected float-pH data in response to the reference pressure choice, the reference depth selection as well as the choice of the method used to correct float cycles. We have demonstrated in the manuscript that significant differences ranging between ca. 0.003 pH units and ca. 0.04 pH units are observed between the four reference methods which can be used to correct float-pH data. In the studied area, this study also shows that differences related to the reference pressure level choice ranged between 0.0047 and 0.0141 pH units (Fig. 3B). By combining the observed possible sources of uncertainty, the
* * *
[11] Newton J.A., Feely R. A., Jewett E. B., Williamson P. and Mathis J. (2015). Global Ocean Acidification Observing Network: Requirements and Governance Plan, Second Edition, GOA-ON, http://www.goa-on.org/docs/GOA-ON.

corresponding uncertainty is either at the edge or well beyond the weather and climate goals, respectively. However, we agree with the referee that several sentences in the original manuscript were out of the scope of this discussion as they were related to accuracy and errors, which is not the purpose of these GOA-ON goals. To clarify the situation, some sentences have been removed and Section 3.3 in the revised manuscript has been shortened and modified. Figure 8 has been also removed in the revised manuscript to clarify it.

379: The TA uncertainty is also far more important if we become interested in DIC.

We agree with the referee that TA uncertainty can lead to noticeable uncertainties when this parameter is used in association to pH data to derive DIC. As stated by Millero (2007)[12], the estimated probable error is even higher when TA values are used in association with pH data to derive DIC ($\pm$ 3.8 mol kg$^{-1}$) than $f$CO$_2$ ($\pm$ 2.1 µatm). However, in the context of converting surface ocean pH measurements into $p$CO$_2$ data for the purpose to derive air-sea CO$_2$ fluxes and determine the ocean behavior with regard to the current atmospheric CO$_2$ increase, Section 3.3. focuses more on this parameter rather than on all the parameters of the marine CO$_2$ system. In the revised manuscript, the sentence has been slightly modified to follow the referee's comment but no in-depth description of the TA uncertainties implications is done.

L. 430: "*This perhaps warrants specific tests on the accuracy of TA predictions in critical regions (or seasons) but also if this parameter aims to be used to derive other parameters of the CO$_2$ system, especially DIC.*"

385: The situation becomes worse still when including uncertainties in the carbonate system constants (Orr et al., 2018)… however, again, these additional uncertainties should be relatively consistent over time, and "weather" and "climate" relate to precision. Many of the concerns raised herein (and the concern over carbonate constants) fall away when you begin to examine variability in the float observations over time

To frame its weather and climate goals, GOA-ON has proposed relative uncertainties thresholds in calculated carbonate ion concentration. These thresholds have been used to back-calculate the corresponding maximum permissible uncertainties in measured input variables. For parameters of the CO$_2$ systems, there are uncertainty contributions from different sources: the instrumental precision, the data conversion uncertainty, and the carbonate system equilibrium constant uncertainties. In consequence, uncertainty propagation should include all those identified sources of bias as it is stated by Orr et al. (2018). In the revised manuscript, this information has been added to better depict the overall concern when deriving $p$CO$_2$ values from float-pH data and TA estimates.

L. 432: "*Finally, an additional source of uncertainty when calculating pCO$_2$ (pH, TA) from floats originates from uncertainties in the carbonate system equilibrium constants (Orr et al., 2018).*"
* * *
[12] Millero, F. J. (2007). The marine inorganic carbon cycle. Chem. Rev., 107 (2), 308-341. https://doi.org/101021/cr0503557.

---

## Author Comment (AC2)

**Technical note: Enhancement of float-pH data quality control methods: A study case in the subpolar northwest Atlantic Ocean**

*In this response, the "original manuscript" refers to the first submitted manuscript that has been evaluated by the reviewer and the "revised manuscript" refers to the manuscript that has been modified according to the reviewer´s comments. Comments from the reviewer are pasted below in black font; our point-by-point responses immediately follow in blue font. Blue italic sentences are those that have been modified / added in the revised manuscript. When indicated, line numbers refer to the new version of the manuscript (Pdf version). We have also submitted a Word document with the "track changes" function activated, which should help the reviewer in figuring out the changes.*

**Responses to REVIEWER 2 - Anonymous Referee**

This paper explores the impact of different calibration choices on the resulting data accuracy for pH data collected by autonomous sensors on Argo floats in the NW Atlantic including comparisons with independent data in the region. The presentation is detailed and will be of high interest to groups seeking to quality control and use data from pH sensors on floats and other autonomous vehicles. The recommendations for choices to make and issues that remain to be solved are helpful. I have a few suggestions that I think will improve the paper and enhance how useful it is to the broader community.

We acknowledge the referee for his/her constructive and interesting comments about our manuscript and for reviewing it. We thank his/her positive opinion about our work and we greatly appreciate that the referee found our work detailed and helpful for the community. In the revised manuscript, all comments and suggestions made by Reviewer #2 have been fully addressed.

*Moderate comments*

Breakpoint nodes: The existence and use of breakpoint nodes in pH sensor calibration may be well known in the pH community already but not to researchers focusing on other sensors. Some discussion of this phenomenon and its likely causes would help this paper be useful to a broader community who may be coming to pH sensor calibration from other sensors. What is the likely cause of these sudden jumps? Discussion of their likely cause would help support the authors' recommendation that the jumps should be gradual rather than sudden.

In particular, I think it would be useful to demonstrate that the sudden jumps are directly related to changing sensor response rather than to apparent discontinuities in the reference value derived from CANYON-B or LIR.

The authors states that these swings in the record are mostly interpreted as changes occurring in the sensor, but this could be supported with the available data. Figure A1 may be partly showing this, but I was unsure if the parking depth pH data had already been corrected or was

raw. A figure (perhaps supplemental) showing the raw pH values and the CANYON-B and LIR values (on different scales) at the relevant pressures would make which signal contained the apparent discontinuities clear.

While drifts and offsets for pH sensors often vary linearly over a long time as for conductivity sensors (Owens and Wong, 2009)[1], calibration jumps are oftentimes observed in the time series, especially over the first few cycles. In the revised manuscript, the reason behind this splitting of the dataset is now more clearly stated "*Indeed, by breaking the time-series sensor record into different segments and fitting each with a linear rate of change in $k_0$, the adjustment better represents the sensor behavior over time as both drifts and offsets change independently between segments and oftentimes noticeable jumps occur over the first few cycles in a float's life (Maurer et al., 2021).*" (L. 186)

This observation has been also reported for nitrate sensors and can be explained by the optics of the nitrate sensor itself which are more sensitive to transient perturbations induced by biofouling. For pH sensors, the sensor $k_0$ value may drift as the sensor warms up and final equilibration with $Br^-$ occurs. The recorded data may also undergo offsets as the system is purged of air bubbles (Johnson et al., 2016)[2]. Thus, Sea-Bird Scientific claims an initial stated accuracy of $\pm 0.05$ pH and a stability of 0.036 pH/year (Bittig et al., 2019)[3]. In the original manuscript, these changes are discussed in Section 2.3 (L. 171): "*Thus, pH sensors are calibrated in the laboratory using spectrophotometric measurements and are therefore directly related to the laboratory calibration method. Each sensor's pressure and temperature coefficients, needed to compute the in situ pH, are also determined in the laboratory as described in Johnson et al. (2016). When deployed at sea, temperature changes modify the reference potential of the sensor and in return induce a sensor drift as the Nernst slope that transforms sensor potential to pH depends on temperature (Johnson et al., 2016, 2017).*". However, the likely cause of these sudden jumps is not well understood so far and we believe that it is somehow out of the scope of our paper. In this manuscript, the purpose of Section 3.1.3. was to discuss the noticeable step-like changes observed with the current correction procedure (i.e. the SAGE method) and to find the best way to represent the smooth sensor drift over time, as observed when looking at the pH time-series recorded at the parking depth (previous Figure A1 in the Supplementary Material). Indeed, in comparison with the pattern of the cycle-by-cycle correction, the high pH changes of ca. 0.01 pH units observed between linear drift phases with the SAGE method appear to be unrealistic, especially when looking at the pH time-series recorded at the parking depth. In our view, the sensor rather shows undulations in response with smooth and less smooth phases and an adaptation of the current correction procedure could be done to better maximize the smoothness of the corrections and to avoid introducing artificial jumps. However, we agree that explanations were missing in the original manuscript and that the original idea to put Figure A1 in the Appendix was not relevant as it is critical for our argument.

Concerning the discontinuities and Figure A1: Following the recommendation of the two reviewers, Section 3.1.3 has been modified, explanations have been added, and Figure 5 re-drawn: now 4 panels representing differences between raw and corrected float-pH data following the SAGE method (panel

[1] Owens, W. B. and Wong, A. P. S. (2009). An improved calibration method for the drift of the conductivity sensor on autonomous CTD profiling floats 565 by θ–S climatology, Deep Sea Research Part I: Oceanographic Research Papers, 56(3), 450-457. https://doi.org/10.1016/j.dsr.2008.09.008.

[2] Johnson, K. S., Jannasch, H. W., Coletti, L. J., Elrod, V. A., Martz, T. R., Takeshita, Y., Carlson, R. J., and Connery, J. G. (2016). Deep-Sea DuraFET: A Pressure Tolerant pH Sensor Designed for Global Sensor Networks, Analytical Chemistry, 88(6), 3249-3256. https://doi.org/10.1021/acs.analchem.5b04653.

[3] Bittig, H. C., Maurer, T. L., Plant, J. N., Schmechtig, C., Wong, A. P. S., Claustre, H., Trull, T. W., Udaya Bhaskar, T. V. S., Boss, E., Dall'Olmo, G., Organelli, E., Poteau, A., Johnson, K. S., Hanstein, C., Leymarie, E., Le Reste, S., Riser, S. C., Rupan, A. R., Taillandier, V., Thierry, V. and Xing, X. (2019) A BGC-Argo Guide: Planning, Deployment, Data Handling and Usage. Front. Mar. Sci. 6:502, https://doi.org/10.3389/fmars.2019.00502.

A), the GEOMAR methods (panel B), pH data measured at the parking depth (panel C) and pH data measured at the parking depth minus reference (CANYON-B pH data, panel D) are presented. Figure A1 has been modified and is now included in Figure 5 of the revised manuscript. When pH data recorded at the parking depth are plotted, they are uncorrected. This precision has been added in the legend of Figure 5 (in the revised manuscript).

By splitting previous Figure 5 (A and B) into two separate figures, we believe that the new organization of the figures helps the reader to identify the impact of the sensor drift correction used on the final adjustment. While the impact of the correction method on the final corrected dataset is almost non-significant, this section aims to discuss the better representation of the sensor behavior over time and we believe that, by merging Figure A1 (in the original manuscript) to the original Figure 5 in revised manuscript, this purpose has been clarified to the reader. We also thank the referee for his/her suggestion to discuss the pH reference values in order to demonstrate that the jumps are related to the sensor response and the correction method. The new Figure 5D showing the difference between uncorrected pH data recorded at the parking depth and pH values from the algorithm estimate is in agreement with this statement as it shows smoothed transitions and the general pattern does not present noteworthy jumps. It confirms that the sudden jumps are related to changing sensor response rather than to the reference values used to correct float-pH data.

L.327: *"The pH sensor behavior when the float drifts at its parking depth is in agreement with this observation (Fig. 5C). In comparison with float-pH data corrected using the SAGE method, no strong visible discontinuities in raw pH data are observed while the float drifts between its measurement's phases. In our view, the sensor rather shows undulations in response with smooth and less smooth phases over time. In order to test the impact of the reference method on the adjustment pattern, differences between uncorrected float-pH data and CANYON-B pH data derived at the parking depth are presented in Figure 5D. Once again, the pH time-series shows smoothed transitions and the general pattern doesn't present noteworthy jumps. Such sharp transitions can perhaps be best corrected with our modified GEOMAR segment method or alternatively with a spline fit or a 3-point centered running mean (Fig. 5B)."*

[Figure]

**Figure 1** (Fig. 5 in the revised manuscript). Mean differences between raw float-pH data minus float-pH corrected using the SAGE tool (Fig.5A), the cycle-by-cycle GEOMAR method (yellow dots, Fig.5B), and the linear mean regression

GEOMAR method (blue dots, Fig.5B) and the 3-point centered running mean correction method (green dots, Fig.5B) for float WMO 3901669. In every case, CANYON-B was chosen as a reference method, and 1500 dbar was chosen as the reference depth. Mean differences between raw and corrected float-pH data with the standard deviations are shown in the legend boxes for each reference method. Figure C shows, for comparison with the SAGE correction, the uncorrected pH data measured at the parking depth (right y-axis) with black dots representing mean pH values for each day. The colorbar shows pressure. Figure D shows differences between raw float-pH data minus float-pH corrected using the SAGE tool (purple dots, left y-axis) and differences between uncorrected mean pH data measured at the parking depth minus mean reference CANYON-B pH data calculated using measurements recorded at the parking depth (red dots, right y-axis).

Section 3.2.1: The authors make the point that the crossover between the float and discrete data is extremely close in time and space. I suggest that they compare temperature/salinity between the float and discrete profiles. This would add evidence that the same water masses were sampled by both.

We thank the referee for his/her suggestion. In the revised manuscript, a new figure has been added in the Supplementary Material (Figure 2 below) to support our comparison between float-pH data and discrete pH measurements. A sentence referring to this figure has been also added in the revised manuscript.

[Figure]

**Figure 2** (Fig. A1 in the revised manuscript)**.** (A and B) Vertical profiles of temperature (A) and salinity (B) measured during the MSM94 cruise (diamond dots) and acquired by the float WMO 3901669 during cycles 121, 122 and 123 (gray and black lines, respectively). (C and D) Differences between discrete and float (cycle 122) temperature (C) and salinity (D) data calculated after matching in density space to avoid biases from internal waves.

Lines 356: "*Differences between discrete and float temperature and salinity data add confidence in the density space matching performed in this study (Fig. A1).*"

First paragraph of section 3.1.1: Very interesting discussion, but I think you could be clearer about whether the main reason for the big offsets between using 1500 and 1960 debar as the reference depth in this region is due to the occasional large convection depths or just because there are offsets in the reference pH values between these two depths, i.e., depth based accuracy differences in CANYON-B or LIR.

The correction scheme for float-pH data involves a comparison of raw float-pH data against selected reference fields where concentrations are relatively stable and can be predicted using interpolation methods (multiple linear regressions or neural networks) based on shipboard observations. In the subpolar North Atlantic Ocean, inter-annually varying deep convection, water mass formation, as well as decadal variability affect water masses at a depth greater than 1500 dbar. The latter shows that the deep ocean correction at an arbitrarily chosen depth around 1500 dbar is not straightforward, and in this case, induced an uncertainty of at least 0.005 pH units. This may to some extent be a special characteristic of the Labrador Sea with its extreme winter convection depth of up to 1500 m. In this particular case, this speaks for a deeper reference level as: (1) raw float-pH data at this perturbed reference might be biases by some units and could not be used to correct the entire profile and, (2) the reference pH values are calculated using physical and biogeochemical properties at the selected pressure range and might change depending on the choice of the reference depth. In the revised manuscript, this statement is now more clearly expressed: "*The deepest mixed layer depth estimated from the float time-series was at 1937 dbar, showing that the entire water column covered by the float profiles is probably affected. In this regard, the subpolar North Atlantic region with its deep-reaching anthropogenic $CO_2$ imprint is certainly a most difficult area for the unambiguous choice of a stable and unperturbed reference depth as both float-pH data and reference pH values could vary noticeably at the classical reference depth.*" (L.238)

We also would like to point out that, in the revised manuscript, a presentation of pH data estimates with the ESPERs routines has been included.

L.284: "*The new ESPERs methods attempt to resolve the issues encountered with existing routines (especially the OA estimate) by expanding their functionality and being trained on a larger data product. In comparison with the LIR-pH estimates, large differences are observed in the SNWA region and might be attributable to the OA adjustment as well as the omission of depth as a predictor variable from ESPER-LIR (Carter et al., 2021). Updated global algorithms (i.e., ESPERs) show comparable estimates in the SNWA area with ESPER-LIR pH estimates slightly higher than pH data estimated with CANYON-B or ESPER-NN. In the dynamic and strongly human-impacted studied region, the lack of coordinate information as a predictor variable in the ESPER-LIR routine could also be argued as an explanation of the observed differences. However, according to Carter et al. (2021), regional assessment statistics obtained in the Northern Atlantic indicate almost similar biases for both the ESPERs and the CANYON-B methods, with a better RMSE statistic for CANYON-B.*"

Line 125: That deployment cast pH measurements are an ineffective means of calibrating float pH sensors seems very important to me. Suggest that you support this statement more strongly. The two papers cited are on oxygen optodes rather than pH sensors. How large a drift rate is typically observed at the start of deployment? When does this change? Could this be overcome with a conditioning period before deployment?

We thank the referee for his/her useful comment and we agree that, in the original manuscript, information about pH sensor behavior during the first cycles were missing.

Recently, Maurer et al. (2021)[6] tested the sensor performance upon deployment by considering the offsets associated with the first and second segments. Indeed, as each segment is treated independently, the value of any subsequent offset can provide information on sensor health over time when viewed relative to the first offset. In the Southern Ocean, a bias of -0.032 pH units has been observed for the first segment. However, similar to nitrate sensors, pH sensors are relatively stable after a few cycles and the second offset was null. The negative skew of the pH first offset distribution demonstrates that the majority of SOCCOM pH sensors are biased low upon deployment within the array. Maurer et al. (2021)[6] stated that this behavior is not surprising; oftentimes the largest anomaly is observed on the first cycle as the sensor re-conditions to an aqueous environment. Continued exposure to seawater at 1500 m helps to stabilize the sensors, particularly the pH sensor. After float-pH data adjustment, median differences between shipboard pH measurements and float-pH data at the time of deployment ranged between 0.006 pH units (Johnson et al., 2017)[4] and 0.002 pH units (Maurer et al., 2021)[6], respectively. In the current literature (Johnson et al., 2018)[5], it is therefore advised to perform the first delayed-mode quality control and adjustment after at least five cycles.

In the revised manuscript, additional information have been added. "*In the Southern Ocean, Maurer et al. (2021) reported an offset value for the first segment of -0.32 pH units, illustrating the sensor performance upon deployment caused by the lack of conditioning in some of the pH sensors as well as the sensor re-conditions to an aqueous environment. However, after float-pH data adjustments, Johnson et al. (2017) and Maurer et al. (2021) showed median shipboard bottle- minus-float differences of 0.006 pH units and 0.002 pH units, respectively.*" (L. 137)

*Minor comments*

Internal waves: The authors take care to remove the effect of internal waves on the discrete – float data comparison. I suspect that this effect is implicitly removed in the calculation of reference pH at a specific depth by CANYON-B and LIR, but it would be useful to state that explicitly.

Internal waves can create mismatches in water properties at a given depth over time between a hydrographic cast and a float profile. As they exist only when the water body consists of layers of different densities, a matching in density space has been performed in our study to allow the comparison between discrete pH measurements and float-pH data. In the current float-pH data adjustment procedure, detailed in Maurer et al. (2021)[6], the difference between a selected field of reference and the measured values is first calculated at nearly 1500 m where spatiotemporal variability of oceanic components is minimal. Thus, the correction scheme for sensor data depends on having accurate estimates of deep chemical concentrations, where concentrations are relatively stable and can be

[4] Johnson, K. S., Plant, J. N., Coletti, L. J., Jannasch, H. W., Sakamoto, C. M., Riser, S. C., et al. (2017). Biogeochemical sensor performance in the SOCCOM profiling float array. J. Geophys. Res. Oceans 122, 6416–6436. https://doi.org/10.1002/2017JC012838.

[5] Johnson, K. S., Plant, J. N., and Maurer, T. L. (2018). Processing BGC-Argo pH data at the DAC level, v1.0, Argo data management, https://doi.org/10.13155/57195.

[6] Maurer, T. L., Plant, J. N., and Johnson, K. S. (2021) Delayed-Mode Quality Control of Oxygen, Nitrate, and pH Data on SOCCOM Biogeochemical Profiling Floats, Frontiers in Marine Sciences, 8, 683207. https://doi.org/10.3389/fmars.2021.683207.

predicted using interpolation methods (multiple linear regressions or neural networks) based on shipboard observations.

Line 60: suggest changing "qualification" to "quality control".

The word has been modified as suggested.

Line 62: suggest adding citations for the suggested procedures.

Citations have been added in the revised manuscript.

L. 67: "*For pH, numerous delayed-mode procedures have been suggested (Williams et al., 2016; Johnson et al., 2017) but a uniform, fully tested and globally-proven correction method is still missing.*"

Line 99: suggest rephrasing. Your paper is evaluating the accuracy range after data adjustment.

This sentence has been rephrased as suggested: "*The initial pH accuracy "claimed" by the manufacturer is of $\pm 0.05$ pH units. Data adjustment can bring accuracies varying between $\pm 0.005$ pH units (Johnson et al., 2017) and $\pm 0.007$ pH units (Maurer et al., 2021).*" (L.107)

Line 106: suggest adding a few more details. Were the adjustments based on air-calibration or surface climatologies or …?

Two references as well as some information about the adjustment have been added in the revised manuscript.

"*In our case, $O_2$ from the 10 pH-equipped Argo floats was adjusted following Argo procedures (Bittig et al., 2018a; Thierry et al., 2022) with in-air measurements and the adjustments are available in near-real time.*" (L. 116)

Line 200: "not" should be "note"

The sentence has been modified.

Lines 246-247: suggest clarifying what is meant by "keep the adjustment". I wasn't sure if you meant that you did apply the optional CANYON-B pH data adjustment to align with spectrophotometric data or you did not.

Carter et al. (2018)[7] have proposed an optional CANYON-B pH data adjustment that can be applied to data derived using this neural network method to align estimates with spectrophotometric pH measurements. As the LIR-pH training dataset consists of values either measured or calculated but adjusted using the same purified-dye adjustment (Equation 1; Carter et al., 2018[7]), we have decided to keep this adjustment to then compare the corrected datasets. In the revised manuscript, this is now more clearly stated: "*In this study, we have decided to include this reference-pH data adjustment to correct*
* * *
[7] Carter, B. R., Feely, R. A., Williams, N. L., Dickson, A. G., Fong, M. B., and Takeshita, Y. (2018). Updated methods for global locally interpolated estimation of alkalinity, pH, and nitrate, Limnology and Oceanography: Methods, 16(2), 119-131, https://doi.org/10.1002/lom3.10232.

*float-pH data: a linear transformation was applied to CANYON-B pH estimates to bring estimates back into alignment with spectrophotometrically measured pH.*" (L. 266)

Line 332: I think "as" should be "than"

The word has been changed.

Line 394: remove comma after both

The comma has been removed.

Figure 6d: Add y-axis labels and negative signs to all y-axis labels.

We thank the referee for pointing out that Figure 6D was incomplete in the original manuscript. In the revised manuscript, Figure 6 (Page 16 in the revised manuscript) has been re-drawn, reduced and now presents differences between discrete pH measurements and float-pH data along the water column and according to two distinct reference pressure levels (1500 dbar Fig. 6A and 1950 dbar Fig. 6B) and ΔpH (discrete pH measurements minus float-pH data corrected at the reference depth level 1950 dbar, Fig. 6C) as a function of the difference between discrete water temperature and temperature values recorded at the reference depth of 1950 dbar (i.e. 3.3733°C). On Fig. 6C, y-axis labels have been corrected.

[Figure]

**Figure 3** (Fig.6 in the revised manuscript). (A and B) Differences between discrete and float-pH data (for the cycle 122) calculated after matching in density space to avoid biases from internal waves and corrected using corrected reference levels of 1500 dbar (Fig. 6A) and 1950 dbar (Fig. 6B). (C) ΔpH (discrete pH measurements minus float-pH data corrected at the reference depth level 1950 dbar) as a function of the difference between discrete water temperature (i.e., the temperature measured in situ at the time of bottle triggering at sea) and temperature values recorded at the reference depth of 1950 dbar. The color code refers to the reference method used to correct float-pH data: CANYON-B (yellow diamonds), LIR-pH (green diamonds), ESPER-NN (purple diamonds) or ESPER-LIR (blue diamonds).

---

## Author Response (AR3)

**Technical note: Assessment of float-pH data quality control methods - A study case in the subpolar northwest Atlantic Ocean**

**Responses to REVIEWER 1 - Brendan CARTER**

Dear authors,

I have gone through the revised version of your manuscript. I commend your hard work on this revised draft and your detailed responses to reviewers. You have attempted to answer all of the comments and questions and many have been addressed satisfactorily. However, I still have concerns about a few aspects:

We appreciate the time and effort that the reviewer dedicated to providing feedback on our manuscript and are grateful for the insightful comments on and valuable improvements to our paper. In the revised manuscript, we have incorporated most of the suggestions made by the reviewer and tried to address his remaining concerns in the following response. In this response, the "original manuscript" refers to the first submitted manuscript that has been evaluated by the reviewer, and the "revised manuscript" refers to the manuscript that has been modified according to the reviewer´s comments. Line numbers correspond to the PDF file.

1. I still have strong misgivings about the mismatch in the salinity. I'm not sure that the central comparison in this paper can be valid with that large of a discrepancy. These appear to be different water masses.

In order to yield a larger number of crossovers, a rather large search window was applied. These crossovers rarely are perfect matches in T and S. Therefore, under the assumption that differences in pH to a major extent are driven by differences in temperature, the $\Delta$pH at $\Delta T = 0$ was calculated. By fitting a linear regression to the data (as the intercept of the regression equation), the pH offset was estimated following the assumption that this regression using crossovers achieved with a relatively wide search window yields a more robust $\Delta$pH estimate as an average of a much smaller number of crossovers found with a smaller search window. We note that by calculating the pH offset as a function of $\Delta S$ (i.e., @ $\Delta S = 0$), the resulting $\Delta$pH values are statistically indistinguishable from the ones based on $\Delta T$ (but have slightly larger uncertainty). This in essence means that the result does not depend significantly on whether we calculate the offset for isothermal ($\Delta T = 0$) or ioshaline ($\Delta S = 0$) conditions. The fact that the isothermal condition is not exactly isohaline (and vice versa) indicates that we do not have a perfect match in water masses. Given the slope of the $\Delta$pH vs. $\Delta S$ regression (Table 1 below and in the revised manuscript), this mismatch seems to not introduce any discernible uncertainty in the pH offset. We therefore based the estimation on the linear regression in T space which shows a moderate sensitivity of $\Delta$pH with $\Delta T$. In consequence, we believe that, by increasing the research window to find crossovers only the uncertainty ($\pm$) is affected but not the mean pH offset.

Nevertheless, given the limitation of our dataset containing only 2 floats, 4 additional floats (that were not part of our pilot study) with trajectories overlapping the SOOP line transect were used to test our assumptions. The ESPER_LIR reference method was used to correct all float-pH data and only float data flagged as "good" were used for this analysis. Except for the floats WMO 1902303 and 1902304

that were corrected using a reference depth around 950 db (940-980 db) as some cycles did not go deeper than 1000 db, all the floats were corrected at 1950 db. Fig. 1A shows that differences between these floats and SOOP-based pH observations (corrected to the temperature of the respective float surface pH observations) do not show a temporal bias, indicating that the SOOP-pH dataset is not biased by a drift pattern; in agreement with Figure 2 below. Moreover, the range of apparent offsets of $\pm 0.03$ pH observed for this 6-float dataset (Fig. 1E) is essentially independent of the search radius criterion, indicating that the large crossover search window does not introduce a bias but only adds more data points (Fig. 1B). Following the comment and suggestion of the reviewer, offsets between SOOP-pH and float-pH data as a function of temperature difference were plotted introducing an additional criterion ($\Delta S < 0.5$) for the cross selection. We note, however, that no significant difference between the calculated pH offsets based on the two different S criteria ($\Delta S < 0.5$, $\Delta S < 2.0$) is observed. As shown in Figure 1D, no slope in the $\Delta$pH vs. $\Delta S$ regression can be reported, highlighting the lack of dependence with this parameter with data linearly spanned in comparison with the temperature-pH dependence (Fig. 1C), confirming our assumption that differences in pH to a major extent are driven by differences in temperature. The pH offset was determined at $\Delta T = 0$ °C (temperature difference between float data and SOOP data) by fitting a linear regression to the data for the float having spread $\Delta T$ values or by considering the mean pH difference when $\Delta T = 0$ (Fig. 1E). Table 1 shows the pH offsets and their uncertainties for the six floats considered and derived using either $\Delta T=0$ or $\Delta S=0$. While the crossovers identified for the six floats are not a perfect match, they all point toward unacceptably high and not constant biases in surface pH values that are too large to be applied as correction.

In the revised manuscript, a reduced $\Delta S$ value of 0.5 is now used to derive the pH offset at $\Delta T = 0$ and this additional comparison plot and table have been added in the Supplementary Material and discussed in the main manuscript.

*"While we found no dependence between $\Delta$pH and $\Delta S$, an additional criterion of $\Delta S \leq 0.5$ has been applied to the crossovers selection in order to exclude major water mass discrepancies."* (L. 401)

*"Calculating the apparent pH offset as a function of $\Delta S$ (Table A2) yields $\Delta$pH values which are statistically indistinguishable from the ones based on $\Delta T$."* (L.415)

*"An extended crossover comparison with the addition of four floats (that were not part of our pilot study) yields mean pH offsets that fall in the range. $\pm 0.03$ pH units (Fig. A2). These mean pH differences are randomly distributed in space and time, indicating an incomplete float-pH data adjustment rather than a drift in the SOOP-reference dataset."* (L. 420)

[Figure]

**Figure 1** (Only Figs. 1A, 1B, and 1E are in the revised manuscript). Offsets between SOOP-pH and fully corrected float-pH data (y-axis) as a function of the time (Fig. 1A), the crossover criterion (Fig. 1B), the temperature difference (SOOP minus float temperature; Fig. 1C), and the salinity difference (SOOP minus float salinity; Fig. 1D). Figure 1E shows the mean offsets and their associated uncertainties for the 6 floats considered. The pH offset was determined at $\Delta T = 0$ °C (temperature difference between float data and SOOP data) by fitting a linear regression to the data for the float having spread $\Delta T$ values (dots) or by considering the mean pH difference when $\Delta T = 0$ (crosses; Figure 1E). Crossovers were calculated for $\Delta x \leq 400$ km, $\Delta t \leq 7$ d, and $\Delta S \leq 0.5$. pH values were recalculated using CO2SYS (van Heuven et al., 2011) to account for any temperature difference between matched observations. Float-pH data have been corrected with the SAGE tool using either the reference depth level 950 dbar or 1950 dbar and ESPER-LIR as reference (see Table 1 below). N stands for the number of values used to derive the statistics.

**Table 1.** (Table A2 in the Supplementary Material of the revised manuscript). Statistics of the crossover analysis for SOOP- and float-pH data. N stands for the number of values used to derive the statistics. Crossovers were calculated for $\Delta x \leq 400$ km, $\Delta t \leq 7$ d, and $\Delta S \leq 0.5$. pH values were recalculated using CO2SYS (van Heuven et al., 2011) to account for any temperature difference between matched observations. Float-pH data have been corrected with the SAGE tool using either the reference depth level 950 dbar or 1950 dbar and ESPER-LIR as reference.

| | | | $\Delta$pH at $\Delta T=0$ | | $\Delta$pH at $\Delta S=0$ | |
|---|---|---|---|---|---|---|
| Correction Depth | N | *Float WMO* | pH offset | Uncertainty of the offset | pH offset | Uncertainty of the offset |
| 950 | 11 | *1902303* | 0.025 | 0.010 | 0.018 | 0.010 |
| 950 | 11 | *1902304* | 0.018 | 0.006 | 0.012 | 0.006 |
| 1950 | 5 | *4903365* | 0.030 | 0.002 | 0.036 | 0.004 |
| 1950 | 2 | *6904241* | 0.042 | 0.0003 | 0.030 | 0.0003 |
| 1950 | 11 | *6904112* | -0.031 | 0.014 | -0.029 | 0.008 |
| 1950 | 8 | *3901669* | -0.004 | 0.007 | 0.013 | 0.006 |

2. I'm not certain that the point was understood about the inherent inaccuracy of pH, and why it's reassuring, but not sufficient for the authors' purposes, that measurements match those of Dickson's lab. If I recall, in this paper the authors are claiming an uncertainty in pH that is better (0.002) than the uncertainty assessed by the Dickson Lab for the measurement technique used to measure seawater reference materials (https://aslopubs.onlinelibrary.wiley.com/doi/abs/10.4319/lom.2013.11.16) (assessed at 0.01 to 0.02 with the potential to improve to as good as 0.005). For calculations of pCO2 from pH, we care about the accuracy of the pH value and not the precision.

During the MSMS94 cruise, total alkalinity, dissolved inorganic carbon, as well as pH measurements, were achieved following the current standard procedure (SOP). During the analysis at GEOMAR, pH measurements were tested regularly against CRM reference samples. While CRMs are certified only for TA and DIC, pH measurements were also performed for each batch by Dickson's lab and made available for us. The comparison done against these certified materials leads us to conclude that the resulting reproducibility in pH measurements for discrete samples was ± 0.002 pH units. We are fully aware that the CRM is not a CRM for pH. However, we think that the Dickson lab is capable of performing accurate pH measurements (with a reported uncertainty of 0.001). With this assumption, and also because all the best practice recommendations have been followed, we are also confident that our SOOP line based pH measurements are accurate and compare well with measurements from a different laboratory (i.e. Dickson lab). Indeed, because of the relatively high stability of the pH measurement, a suite of 5-8 repeated CRM-reference measurements are performed in port before and after each 5-week autonomous roundtrip (Fig. 2 below). These pre- and post- calibration runs are rather stable for each individual meta-cresol purple (mCP) indicator bag (but somewhat different between individual bags). This yields a clear and consistent track of the small pH drift over consecutive roundtrips which allows us to correct the measured pH to CRM values. Given the small standard deviations of the CRM measurements, we believe that the SOOP-pH is of about 0.003 pH units. In the revised manuscript, the term uncertainty has been deleted and replaced by reproducibility as suggested by the referee.

In addition, we note that the pH offsets of the 6 selected floats are in the range +0.04 to -0.03, i.e. span a range of 0.07. So even if there was a bias in the CRM pH, the range would remain exactly the same, only the absolute pH differences would shift accordingly. So, the conclusion that in the subpolar North Atlantic, the current pH cookbook procedures do not yield well enough constrained pH is still valid. Only if all 6 floats produced essentially (within error) the same offset, an accuracy issue on the CRM pH would have to be invoked.

[Figure]

**Figure 2** (not in the revised manuscript). pH measurements performed on CRM batch 190 with the CONTROS HydroFIA pH system before and after each 5-week roundtrip of the DE-SOOP Atlantic Sail. Adjustments of measured pH to the nominal pH value assigned to the CRM ($7.8417 \pm 0.0014$ at 25°C) are based on the mean of all CRM measurements carried out per individual meta-cresol purple bag.

However, we agree with the referee that other sources of uncertainty (i.e., the lab-to-in situ pH conversion uncertainty introduced through calibration (Williams et al., 2017) and the bottle pH inaccuracy) have to be considered when comparing the discrete and float-pH datasets.
L. 375: "*Moreover, the laboratory-to-in-situ temperature pH conversion uncertainty of 0.005 pH units (Williams et al., 2017), as well as the absolute uncertainty in the bottle pH measurements (here 0.002 pH units), have to be taken into account before drawing strong conclusions.*"

Finally, we do agree with the referee that the final pH accuracy is crucial for the calculation of $pCO_2$ from pH. In the manuscript, after discussing the different possible sources of uncertainty, Section 3.2. aims to compare float-pH data against discrete pH measurements to assess the accuracy of the correction procedure and then discuss the errors of the corrected datasets. The following Section 3.3 tends to link the two previous ones by presenting the impacts of the uncertainties associated with the current adjustment procedure on the final accuracy and thus the derived parameters such as the $pCO_2$.

3. I still don't understand why the temperature relationship seems so discrepant from other examples of similar measurements.

In the studied area, two crossover comparisons have been performed using two independent datasets and on two different floats. Despite the limited number of floats and crossovers associated with this study providing only one showcase, and although the actual pH values may be slightly different due to the regional variability, in both cases, pH offsets are positively correlated with temperature, being the smallest at the temperature of the reference depth (Figure 6C in the manuscript). As it is stated in the manuscript, we argue that it points towards an imperfect representation of the temperature and/or pressure dependencies of the pH sensor (Page 17). One possible explanation of that imperfect temperature dependence could be linked to the TCOR ratio used in SAGE to adjust the sensor k0 (that is what is assumed to drift): it is addressed by normalizing the adjustment along the profile to the temperature at which the adjustment was derived. As the temperature gradient could be high along the

water column in this region (> 10°C), we wonder if this TCOR term may induce an under-correction when high temperatures are recorded at the surface and thus an under-representation of seasonal thermal changes. By using either the temperature at 1500 dbar or the temperature at 5 dbar, the TCOR term varies between 0.09618 and 0.9999. For a mean pH value of 7.8273 and an offset of 0.0762 (hypothetical), it represents pH values equal to 7.9006 and 7.9035, respectively, representing a difference of ca. 0.003 pH units.

Also, the pH sensor laboratory-calibration before its deployment could be another possible explanation as pressure and temperature coefficients are determined at this stage. A possible uncorrected or incomplete calibration at this stage could induce biased derived float-pH data. Finally, another possible explanation could be related to the established at-depth correction that does not seem to yield adequate pH accuracy at the surface, at least in the subpolar North Atlantic. This uncertainty may partly be incurred by the regional complication of finding a reliable at-depth reference. Following a suggestion from Reviewer #2 (added in the revised manuscript), differences between raw minus corrected float-pH data have been calculated for winter cycles during which the MLD was deeper than 1000 dbar: differences are larger when the classical reference depth is used. Conversely, lower deviations between raw and corrected pH data are measured when the deepest reference depth is used. In this area, we believe that this speaks for a deeper reference level to corrected float-pH data as late winter cycles are more prone to be perturbed at the classical reference depth and thus could not be used to correct the entire profile. An improved understanding of the temperature and pressure effects on the sensor could be a way forward to improve float-pH data adjustment.

4. Despite limiting the data used slightly, I still think that the standard deviation is presented in an unhelpful way. You have a large standard deviation relative to the interquartile ranges and even the 95th percentiles, and as pointed out that is because of a small number of extreme outliers. This is problematic because you are actually applying some of those estimation relationships in locations where those relationships are explicitly not intended to work. Thus, the resulting standard deviation is not useful and might actually confuse readers as it did me.

We agree that we could have been stricter on removing questionable regions in our comparison for the previous revision, and have done so now by removing both Baffin Bay (little effect) and the Mediterranean Sea (noticeable reduction of the standard deviation) entirely, in addition to the High Arctic and Black Sea as done previously. In the revised manuscript, Figure 4 has been modified accordingly (Page 13).

We see the point that the reviewer makes here on the mismatch between a larger-than-expected and thus confusing standard deviation compared to expectations from the 5th/95th percentiles, but we also do see merit in stating exactly this mismatch. Indeed, this is because of a 'smallish' number of data areas that do not conform to the expected normal distribution behavior. However, as can be seen from the color scale on the delta pH maps (and more prominently in the modified Figure 3 below where only data outside the 5th/95th percentiles are retained), these "outliers" largely lie in open ocean locations (such as the North Atlantic, the Indian Ocean, or the North and Tropical Pacific) where those estimation relations are indeed intended to work, but give noticeable ('extreme' in terms of desired pH accuracy) differences. We would then argue to keep the standard deviation for the above purpose.

[Figure]

**Figure 3** (Fig. 4 in the revised manuscript). Spatial distributions of estimated pH data at the classical reference depth 1500 m using different reference models: LIR-pH (with the OA adjustment) (a), CANYON-B (f), ESPER-LIR (k), and ESPER-NN (p). Maps of the spatial difference between the estimated pH datasets are presented in panels (b-d, g-h and i). Panels (e, i-j and m-o) show the bias ΔpH distribution only for data outside the 5th/95th percentiles (with statistics). The upper colorbar indicates the difference between estimated pH data using the different models and the lower colorbar gives the pH values. For clarity, pH data estimated for the Black Sea, the Baffin Bay, the Mediterranean Sea, and the High Arctic have been removed for this simulation as they were outside the 5th/95th percentile and they caused a noticeable increase of the standard deviation (std).

We also would like to point out that, in response to Reviewer #2, the dataset used as the input variable is also now mentioned in the revised manuscript (World Ocean Atlas climatology). Indeed, Reviewer#2 asked a related question on the same point that has been addressed in his/her response letter.

**Responses to REVIEWER 2 - Anonymous Referee**

This paper uses data from two floats to point out several potential sources of bias arising from current Argo float pH calibration methods: choice of reference depth, choice of reference algorithm, fitting of the correction into straight segments with large discontinuities, and uncertainties in depth or pressure dependence of the correction. These are all important issues and so the paper makes an important contribution by demonstrating their impact on real float data in an important region for understanding ocean carbon. The paper only seeks to offer a recommendation for the treatment of breakpoints in the correction, but is still a useful contribution despite not proposing specific solutions for each issue. I have a few additional suggestions (1 new one – sorry, that I didn't notice this potential problem during my previous review! 2 other moderate ones that follow up on my previous recommendations).

We appreciate the time and effort Reviewer 2 spent on our paper. His/her thoughtful comments and suggestions have helped improve it and we would like to express our thanks and appreciation to Reviewer 2 in these over-committed nowadays. We have addressed all of the comments and included responses in italics below each reviewer's comments. In this response, the "original manuscript" refers to the first submitted manuscript that has been evaluated by the reviewer, and the "revised manuscript" refers to the manuscript that has been modified according to the reviewer´s comments. Line numbers correspond to the PDF file.

**Moderate comments**

Potential problems when using oxygen data from > 1900 dbar in calculating reference pH values: A possible issue exists in calculating the reference pH values at 1950 dbar from algorithms that use O2 data. A frequent problem exists in deep O2 data from Argo floats where the first few, deepest points of a profile are biased low, the so-called "hook". To my knowledge, the origin of this low bias is unknown, and it doesn't affect every profile. However, the authors should confirm whether this bias exists in some of their profiles and remove it before calculating reference pH. The offset shown in Figure 3 appears to be in the correct direction to be caused by 1950 dbar O2 data that is biased low, though I'm unsure at what magnitude the bias could affect the pH data given that typical hook bias is less than 10 umol-O2/kg. Also, suggest adding information about which reference estimation algorithms require O2 data. CANYON-B is mentioned for this in Line 112. However, ESPER, in particular, appears to be a family of possible algorithms depending on what input data is available, so which equation is being used should be specified.

We agree with the referee that, although Argo floats typically measure up to 2000 meters, an unlikely "hook" in the oxygen data at the deepest 50 meters trending toward low oxygen values is observed for several profiles of some Argo floats. Although the main cause is still being investigated, it has been proposed that these 'hooks' are either from optode response time or bio/particle fouling[1]. However, for the three Argo floats considered in our study (WMOs 3901668, 3901669, and 690412), such a bias has not been observed on the oxygen profiles (Figure 1). In consequence, we believe that this bias does not impact our dataset. In the revised manuscript, this statement has been added in Section 2.1.
* * *
[1]Wolf, M. K. (2017). Oxygen saturation surrounding deep-water formation events in the Labrador Sea from Argo-O2 data, (Master's thesis). Retrieved from [UVicSpace]. (https://dspace.library.uvic.ca//handle/1828/8401). Victoria, BC: University of Victoria.

[Figure]

**Figure 1** (not in the revised manuscript). Oxygen concentration profiles for (A & D) the entire water column, (B & E) the depth comprised between 1450 and 1550 dbar and (C & F) the bottom 1900 to 2000 db for floats 3901668 (upper panels) and 3901669 (lower panels). The low oxygen 'hook' is not visible on any profile.

In order to test the impact of an oxygen change of 10 µmol kg$^{-1}$ on the estimated pH values, mean temperature, salinity, and oxygen values measured at 1500 db and 1950 db for the float 3901669, as an example, were used to compute CANYON-B pH values (Table 1).

**Table 1** (Not of the revised manuscript). Parameter values used as inputs to determine the pH changes corresponding to an oxygen change of 10 µmol kg$^{-1}$. Computation has been done using the CO2SYS software.

| Mean Temp. [°C] | Mean Sal. | Mean. Oxygen [µmol/kg] | Mean Depth (db) | Mean pH | Mean pH (+ 10 µmol/kg of Oxygen) | pH difference |
|---|---|---|---|---|---|---|
| 3.3827 | 34.8839 | 277.3410 | 1500 | 7.9726 | 7.9784 | **± 0.0058** |
| 3.3780 | 34.9253 | 271.6621 | 1950 | 7.9626 | 7.9678 | **± 0.0052** |

Concerning these results, we agree with the referee that pH estimates are sensitive to other input parameters, including oxygen data. The difference between pH data estimated with oxygen values varying between ± 10 µmol kg$^{-1}$ ranges between ±0.0052 to ±0.0058 pH units (with CANYON-B), well below the observed difference reported in Figure 3A (in the manuscript). Moreover, when the same oxygen value as the one measured at 1500 db (i.e., 277.3410 µmol kg$^{-1}$) is used to derive pH data using the mean temperature and salinity values recorded at 1950 db, a pH value of 7.9655 is obtained, highlighting the high sensitivity of this reference algorithm to other input parameters. In this regard, we

would like to point out that the choice of a deeper reference depth tends to use data from a stable and unperturbed reference depth. By considering a negative bias for the oxygen data measured at 1950 db, as it could happen in a "hooked" oxygen data situation, the difference between pH data corrected at 1500 db minus float-pH data corrected at $1950 + 10$ µmol kg$^{-1}$ is equal to 0.0048 pH units (i.e., 7.9726 - 7.9678), still below the difference reported on Figure 3A. In consequence, we do not believe that the difference reported between the two corrected datasets could rely on biased oxygen-deep values. Nevertheless, the oxygen-uncertainty impact on the derived pH values is discussed in the revised manuscript.

*"When O$_2$ sensors incapable of in-air referencing are used (e.g., SBE63 optode, Sea-Bird Electronics), oxygen values typically have uncertainties up to ca. 3% (Takeshita et al., 2013), adding an additional source of uncertainty when these data are used as input parameters to derive reference-pH data."* (L.116)

Finally, we thank the referee for pointing out that algorithm descriptions were incomplete in the original manuscript concerning oxygen data utilization. In this study, all the reference algorithms used employ oxygen values as ancillary data (ESPER methods and LIR regressions #7; Carter et al., 2018, 2021). In the revised manuscript, details have been added.

*"Although some Argo float profiles could be impacted by a "hook" in the oxygen data at the deepest 50 meters inducing low oxygen values (Wolf et al., 2018), a visual inspection of the oxygen profiles from these three floats has been performed and does not point toward such a bias."* (L. 134)

*"In this study, oxygen data were used as predictor variables in all reference algorithms used." (L.112)*

Impact of convection depth on the 1500 dbar correction: In lines 238-246, the authors imply that deeper convection depth in this region is responsible for the offsets between the 1500 dbar reference depth and 1950 dbar reference depth corrections. I think the authors have the data to provide stronger evidence for this implication. If deep convection depths are responsible, late winter cycles where mixed layer is deep should show larger deviations or variability between the raw and reference pH for 1500 dbar. Is that the case for these floats?

In the studied region, deep convection events, water mass formation as well as decadal variability affect water masses at a depth greater than 1500 dbar. By comparing float-pH data corrected at two different depths (i.e., 1500 and 1950 db), this paper highlights that the choice of an arbitrarily chosen depth around 1500 dbar induces an uncertainty of at least 0.005 pH units. Arguing that this result highlights the implication of deep convection events on the current quality control procedure, stronger evidence arises when a comparison between raw pH data minus corrected float-pH data (at the two reference depths) is done (Table 2), as suggested by Reviewer #2. Indeed, when only winter cycles during which the MLD was deeper than 1000 dbar are considered, differences between raw minus corrected float-pH data are larger when the classical reference depth is used. Conversely, lower deviations between raw and corrected pH data are measured when the deepest reference depth is used. In this area, we believe that this speaks for a deeper reference level to corrected float-pH data as late winter cycles are more prone to be perturbed at the classical reference depth and thus could not be used to correct the entire profile. Following the referee's comment, a new Table (Table 2 below) and additional explanations have been added in the Supplementary Material and Section 3.1.1 of the revised manuscript, respectively.

*"By splitting the dataset to keep only profiles done when the MLD was deeper than 1000 dbar, the comparison between raw and corrected float-pH data using the two reference pressures reveals larger variabilities when the classical reference depth is used rather than when the deepest one is considered, highlighting the implication of deep convection events on the adjustment method (Table A1). "* (L. 248)

**Table 2** (Table A1 in the Supplementary Material of the revised manuscript). Mean absolute differences between float-pH data corrected at two distinct depths and using the four different reference methods for the floats WMO 3901668 and 3901669. SD stands for Standard Deviation. Only profiles performed when the MLD was deeper than 1000 dbar were used.

| | | Float WMO 3901668 | | Float WMO 3901669 |
| --- | --- | --- | --- | --- |
| | | *Winter 2019* Mean MLD=1639.6 db | *Winter 2020* Mean MLD=1712.1 db | *Winter 2019* Mean MLD=1240.2 db |
| **ESPER-NN** | *Raw-1500 db* | $-0.0293 \pm 1.39\times10^{-4}$ | $-0.0248 \pm 8.12\times10^{-5}$ | $-0.0352 \pm 9.97\times10^{-5}$ |
| | *Raw-1950 db* | $-0.0181 \pm 5.23\times10^{-4}$ | $-0.0164 \pm 1.08\times10^{-4}$ | $-0.0265 \pm 5.95\times10^{-5}$ |
| **ESPER-LIR** | *Raw-1500 db* | $-0.0416 \pm 3.52\times10^{-5}$ | $-0.0380 \pm 1.64\times10^{-4}$ | $-0.0456 \pm 1.05\times10^{-4}$ |
| | *Raw-1950 db* | $-0.0244 \pm 3.30\times10^{-3}$ | $-0.0300 \pm 1.80\times10^{-3}$ | $-0.0307 \pm 1.77\times10^{-4}$ |
| **CANYON-B** | *Raw-1500 db* | $-0.0508 \pm 8.89\times10^{-5}$ | $-0.0478 \pm 1.41\times10^{-4}$ | $-0.0554 \pm 6.49\times10^{-5}$ |
| | *Raw-1950 db* | $-0.0391 \pm 1.50\times10^{-3}$ | $-0.0399 \pm 8.88\times10^{-4}$ | $-0.0457 \pm 6.20\times10^{-5}$ |
| **LIR-pH** | *Raw-1500 db* | $-0.0677 \pm 4.31\times10^{-5}$ | $-0.0647 \pm 3.95\times10^{-5}$ | $-0.0728 \pm 8.51\times10^{-5}$ |
| | *Raw-1950 db* | $-0.0549 \pm 4.03\times10^{-4}$ | $-0.0543 \pm 2.77\times10^{-4}$ | $-0.0628 \pm 6.18\times10^{-5}$ |

Showing raw and reference values on the same panel: I'm glad to see the information in Figure 5 brought together, but I still think an additional panel or two would clarify the extent to which the breakpoints are caused by sudden changes in raw pH vs. potential discontinuities in the reference pH possibly from spatial patterns as the float moves to different regions. I suggest a panel showing the raw 1500 dbar pH and the reference pH from one or more of the algorithms. Two different scales (like the authors have used in panel c) would allow the signals to be at sufficient vertical resolution. The difference between the raw and corrected is interesting but doesn't provide this crucial information about how stable the reference pH itself is over the spatial movement and timing of the float. In particular, panel c suggests that the breakpoints are either not aligned with discontinuities in the raw data or that the raw data discontinuities at the parking depth are poorly related to those at 1500 dbar. Either way, this would be useful to explore further.

The purpose of this section was to discuss the noticeable step-like changes observed with the current correction procedure (i.e., the SAGE method) and to find the best way to represent the smooth sensor drift over time, as observed when looking at the pH time-series recorded at the parking depth (Fig. 5C). Indeed, in comparison with the pattern of the cycle-by-cycle correction, the high pH changes of ca. 0.01 pH units observed between linear drift phases and leading to step-like changes with the SAGE method appear to be unrealistic (ex. in July 2018). In our view, the sensor rather shows undulations in response with smooth and less smooth phases as it is somehow confirmed by Figure 5C as well as in Figures 5E and F showing raw float-pH data at 1500 dbar.

Indeed, in comparison with float-pH data corrected using the SAGE method, no strong visible discontinuities in raw pH data are observed while the float drifts between its measurement phases and before the application of the pH adjustment procedure. Such smooth transitions can perhaps be best

corrected with our modified GEOMAR segment method or alternatively with a spline fit or a 3-point centered running mean (Fig. 5B). In order to test the impact of the reference method on the adjustment pattern, differences between uncorrected float-pH data and CANYON-B pH data derived at the parking depth are presented in Figure 5D. Once again, the pH time-series shows smoothed transitions and the general pattern doesn't present noteworthy jumps. Nevertheless, we thank the referee for suggesting modifying Figure 5 to better discuss the wiggles around the jumps and their possible link to variability in the algorithm estimates or short-term sensor changes. Indeed, high variability is observed on the reference pH time-series estimated using both CANYON-B (Fig. 5E) or ESPER-LIR (Fig. 5F), highlighting the noticeable impact of the algorithm estimate discontinuities on the final correction while raw float-pH data are not presenting sudden changes. In the revised manuscript, Figure 5 has been modified and some information added to better clarify the impact of reference methods on the breakpoint determination.

[Figure]

**Figure 3** (Figure 5 in the revised manuscript). Differences between raw float-pH data minus float-pH corrected using the SAGE tool (Fig.5A), the cycle-by-cycle GEOMAR method (yellow dots, Fig.5B), and the linear mean regression GEOMAR method (blue dots, Fig.5B) and the 3-point centered running mean correction method (green dots, Fig.5B) for float WMO 3901669. In every case, CANYON-B was chosen as a reference method, and 1500 dbar was chosen as the reference depth. Mean differences between raw and corrected float-pH data with the standard deviations are shown in the legend boxes for each reference method. Figure C shows, for comparison with the SAGE correction, the uncorrected pH data measured at the parking depth (right y-axis) with black dots representing mean pH values for each day. The colorbar shows pressure. Figure D shows differences between raw float-pH data minus float-pH corrected using the SAGE tool (purple dots, left y-axis) and differences between uncorrected mean pH data measured at the parking depth minus mean reference CANYON-B pH data calculated using measurements recorded at the parking depth (red dots, right y-axis). Figures 5E and F show mean raw float-pH data measured at 1500 dbar (between 1480 and 1520 dbar) and pH data calculated by the reference methods CANYON-B (panel E) and ESPER-LIR (panel F) using as input parameters (i.e. temperature, salinity, pressure and oxygen) the values measured

by the float at 1500 dbar. For panels A to D, differences are calculated for each cycle at each depth along the entire profile and then averaged.

"*Moreover, high variability is observed on the reference pH time-series estimated using both CANYON-B (Fig. 5E) or ESPER-LIR (Fig. 5F), highlighting the noticeable impact of the reference algorithms discontinuities on the final correction while raw float-pH data are not presenting sudden changes. Indeed, the raw pH time-series shows smoothed transitions and the general pattern doesn't present noteworthy jumps.*" (L. *334*)

**Minor comments**

Lines 267-268: Suggest giving the mean and standard deviation of this correction at 1500 dbar for this dataset.

This information has been added in the revised manuscript as follows: "*For the two floats considered in this Section, means and standard deviations of the difference between float-pH data corrected at 1500 dbar using CANYON-B and CANYON-B adjusted are equal to 0.0055 $\pm$ 6.63$\times$10$^{-5}$ and 0.0055 $\pm$ 8.31$\times$10$^{-5}$, respectively.*" (L.278)

Line 273: Suggest briefly stating the dataset used as the input variables in the algorithms to generate Figure 4. World Ocean Atlas perhaps?

World Ocean Atlas climatology data was used to create the maps and comparisons of Figure 4. This information has been added to the revised manuscript. It was chosen to cover a reasonable data space like being encountered by profiling floats globally. While individual float profiles may be slightly more accurate examples of reality, their distribution is spotty and doesn't provide the same global coverage as climatology. Besides, differences in algorithms are not caused by climatological vs. real profile input data, but by differences in the algorithms' training data.

Here in the algorithm training data, LIR, CANYON-B, and ESPER show some differences, which could be sources for some of the more extreme differences seen in Figure 4. Both LIR and CANYON-B use the original release of GLODAPv2, however with a different treatment of anthropogenic carbon as well as on pH (from different methods, spectrophotometrically measured or calculated from other CO2 parameters). One could speculate that this is a source for noticeable differences between LIR and CANYON-B (and LIR and ESPER) in convective areas of the North Atlantic. ESPER, in contrast, uses an updated version of GLODAPv2 (GLODAPv2.2020), which includes some modifications on 90's $CO_2$ data in the Indian Ocean. This is a likely cause for the noticeable differences seen between LIR/CANYON-B and ESPER in the Indian Ocean.

Table 2: It's unclear to me why absolute difference is shown here. The sign of the offset seems relevant, and I suggest including it.

Table 3 has been modified in the revised manuscript and the sign of the offset has been added.

| y-x | | 1500 db | | | | 1950 db | | | |
|---|---|---|---|---|---|---|---|---|---|
| | | LIR-pH | CANYON-B | ESPER-LIR | ESPER-NN | LIR-pH | CANYON-B | ESPER-LIR | ESPER-NN |
| WMO 3901668 | LIR-pH | / | 0.0175 ± 0.0012 | 0.0264 ± 0.0026 | 0.0407 ± 0.0028 | / | 0.0155 ± 0.0011 | 0.0259 ± 0.0052 | 0.0399 ± 00027 |
| | CANYON-B | -0.0175 ± 0.0012 | / | 0.0089 ± 0.0016 | 0.0232 ± 0.0019 | -0.0155 ± 0.0011 | / | 0.0105 ± 0.0045 | 0.0245 ± 0.0024 |
| | ESPER-LIR | -0.0264 ± 0.0026 | -0.0089 ± 0.0016 | / | 0.0143 ± 0.0013 | -0.0259 ± 0.0052 | -0.0105 ± 0.0045 | / | 0.0140 ± 0.0053 |
| | ESPER-NN | -0.0407 ± 0.0028 | -0.0232 ± 0.00196 | -0.0143 ± 0.0013 | / | -0.0399 ± 0.0027 | -0.0245 ± 0.0024 | -0.0140 ± 0.0053 | / |
| WMO 3901669 | LIR-pH | / | 0.0173 ± 0.0014 | 0.0321 ± 0.0069 | 0.0391 ± 0.0021 | / | 0.0161 ± 0.0017 | 0.0381 ± 0.0076 | 0.0356 ± 0.0018 |
| | CANYON-B | -0.0173 ± 0.0014 | / | 0.0148 ± 0.0058 | 0.0217 ± 0.0014 | -0.0161 ± 0.0017 | / | 0.0221 ± 0.0070 | 0.0196 ± 0.0014 |
| | ESPER-LIR | -0.0321 ± 0.0069 | -0.0148 ± 0.0058 | / | 0.0069 ± 0.0055 | -0.0381 ± 0.0076 | -0.0221 ± 0.0070 | / | -0.0025 ± 0.0074 |
| | ESPER-NN | -0.0391 ± 0.0021 | -0.0216 ± 0.0021 | -0.0069 ± 0.0055 | / | -0.0356 ± 0.0018 | -0.0196 ± 0.0014 | 0.0025 ± 0.0074 | / |

**Table 3** (Table 2 in the revised manuscript). Mean differences (y-x) between float-pH data corrected at two distinct depths and using the four different reference methods for the floats WMO 3901668 and 3901669. SD stands for Standard Deviation.

Figure 5: The y-axis labels in this figure suggest that they show the absolute value of the raw – reference pH. Here too, I think the sign is important and the y-axis should be just the difference, not the absolute value. The caption is unclear what "mean" difference means. Given that a time series is shown, it's not the mean over the time series.

The y-axis labels have been modified in the revised Figure 5 (see Figure 3 above) and now show raw minus corrected data. In this Figure, we have plotted the mean difference per cycle, i.e. the entire raw profile minus the entire corrected profile is meant. In the revised manuscript, the caption has been clarified.

*"[...] For panels A to D, differences are calculated for each cycle at each depth along the entire profile and then averaged." (Page 15)*